# Local Intrinsic Dimension of Representations Predicts Alignment and Generalization in AI Models and Human Brain

Junjie Yu [* 1 2]   Wenxiao Ma [* 1]   Chen Wei [1]   Jianyu Zhang [1]   Haotian Deng [1]   Zihan Deng [1]   Quanying Liu [1 2 3]

## Abstract

Recent work has found that neural networks with stronger generalization tend to exhibit higher representational alignment with one another across architectures and training paradigms. In this work, we show that models with stronger generalization also align more strongly with human neural activity. Moreover, generalization performance, model–model alignment, and model–brain alignment are all significantly correlated with each other. We further show that these relationships can be explained by a single geometric property of learned representations: the local intrinsic dimension of embeddings. Lower local dimension is consistently associated with stronger model–model alignment, stronger model–brain alignment, and better generalization, whereas global dimension measures fail to capture these effects. Finally, we find that increasing model capacity and training data scale systematically reduces local intrinsic dimension, providing a geometric account of the benefits of scaling. Together, our results identify local intrinsic dimension as a unifying descriptor of representational convergence in artificial and biological systems.

## 1. Introduction

Recent advances in foundation models have revealed two striking phenomena. First, models trained on diverse tasks often converge toward similar internal representations, and this convergence tends to be stronger among models with better generalization performance (Huh et al., 2024; Nguyen et al., 2020). Second, representation of foundation models can align remarkably well with human neural activity, despite being trained without biological supervision (Yamins

et al., 2014; Khaligh-Razavi & Kriegeskorte, 2014; Güçlü & Van Gerven, 2015; Eickenberg et al., 2017; Wang et al., 2023; Conwell et al., 2024; Shen et al., 2024; Raugel et al., 2025). Together, these findings have inspired the Platonic Representation Hypothesis: the idea that successful intelligent systems, both biological and artificial, converge toward a shared representation of the external world (Huh et al., 2024).

However, existing evidence for this hypothesis remains fragmented. Prior work has linked generalization to model–model alignment and has separately demonstrated alignment between artificial models and brain activity (Muttenthaler et al., 2025; Mahner et al., 2025; Huh et al., 2024). Yet these two lines of work typically rely on different model families, datasets, and alignment metrics, making it unclear whether they reflect a common underlying phenomenon. What remains untested is whether these observations form a single correlated structure: **whether models that generalize better also align more strongly with both other models and the human brain**. If the Platonic representation reflects an optimal solution shaped by the structure of the world, we should observe precisely such a systematic relationship.

In this work, we address this question through a large-scale joint analysis of vision models and human fMRI recordings. We evaluate generalization performance, AI–AI alignment, and AI–Brain alignment for the same set of models under a common protocol. This unified analysis reveals that the three quantities systematically co-vary across models: models that generalize better also align more strongly with other models and with human neural activity. These results suggest that representational convergence is not merely an incidental consequence of modern architectures or training objectives, but may reflect shared geometric structure associated with successful generalization.

This observation raises a second question: **what property of representation geometry underlies this convergence?** A geometric answer would provide a more precise characterization of the shared structure toward which artificial and biological systems appear to converge. We propose that intrinsic dimension offers a natural lens for this question. Intrinsic dimension measures the effective number of degrees of freedom needed to describe a representation. Representa-

[1]Department of Biomedical Engineering, Southern University of Science and Technology, Shenzhen, China [2]Omni-Intelligence, Shenzhen, China [3]Shenzhen Loop Area Institute , Shenzhen, China. Correspondence to: Quanying Liu <liuqy@sustech.edu.cn>.

*Proceedings of the $43^{rd}$ International Conference on Machine Learning*, Seoul, South Korea. PMLR 306, 2026. Copyright 2026 by the author(s).

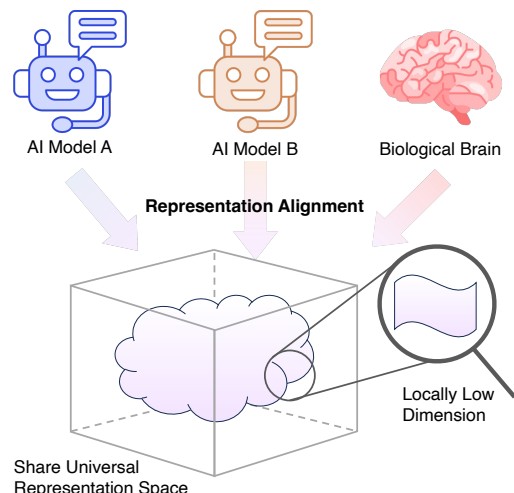

AI Model A     AI Model B     Biological Brain

**Representation Alignment**

Locally Low Dimension

Share Universal Representation Space

*Figure 1.* **Motivation and overview of representational alignment.** Higher-performing AI models exhibit more similar representations to each other and to human neural activity. These convergent representations are also locally low-dimensional, suggesting a simple geometric correlate of cross-model and AI-Brain alignment.

tions with lower intrinsic dimension are more constrained geometrically, whereas representations with higher intrinsic dimension occupy more diffuse regions of embedding space. Prior work has linked lower intrinsic dimension to improved generalization (Ansuini et al., 2019). If better generalization is also associated with stronger AI–AI and AI–Brain alignment, then intrinsic dimension becomes a natural candidate for a common geometric factor underlying all three phenomena. Importantly, it can be estimated directly from embeddings without requiring class labels, making it applicable to both supervised and self-supervised models.

We test this hypothesis by analyzing intrinsic dimension of embedding across models and comparing it with generalization, AI–AI alignment, and AI–Brain alignment. We find that lower-dimensional representations are consistently associated with better generalization and stronger alignment both across artificial systems and with human brain activity. Crucially, these relationships are more strongly associated with *local* intrinsic dimension measured within fine-grained neighborhoods of representation space than with global dimension measures (Figure 1). Finally, we examine how this local geometry changes with scale. We find that increasing model capacity and training data scale systematically reduce local intrinsic dimension, suggesting that scaling progressively refines representations into increasingly shared, low-dimensional local structures. Together, these findings identify local intrinsic dimension as a geometric signature of representational convergence across artificial and biological systems.

Our contributions are threefold:

- **Unified Framework**: We provide the first joint analysis of AI-AI alignment, AI-Brain alignment, and generalization within a single experimental framework, demonstrating that these three quantities systematically co-vary across models.

- **Geometric Descriptor**: Through a multi-scale analysis, we identify local intrinsic dimension as the key geometric correlate of this convergence, a non-trivial finding absent from prior work relying on fixed-scale estimates.

- **Scaling Geometry**: We show that model capacity and training data scale are systematically associated with lower local dimension, providing a geometric perspective on why larger foundation models tend to generalize better and align more strongly with both models and brains.

## 2. Related Work

**AI-Brain Representational Alignment.** Early research established that deep neural networks, particularly those trained on object classification, could predict neural responses in the primate visual system (Yamins et al., 2014; Khaligh-Razavi & Kriegeskorte, 2014; Güçlü & Van Gerven, 2015; Eickenberg et al., 2017; Schrimpf et al., 2021). These findings were often interpreted as a consequence of task-specific supervised optimization, with brain alignment viewed as a byproduct of training on category labels. More recent work has shown that such alignment can also emerge in large-scale foundation models trained without neural supervision (Wang et al., 2023; Conwell et al., 2024; Shen et al., 2024; Raugel et al., 2025; Goldstein et al., 2024; Gokce & Schrimpf, 2025; Raugel et al., 2026). Prior efforts to explain variation in AI-Brain alignment have mainly focused on model-level factors, identifying training objective and data scale as key determinants of alignment (Wang et al., 2023; Conwell et al., 2024; Gokce & Schrimpf, 2025; Raugel et al., 2026). However, they do not explain how these model-level factors affect the geometry of learned embeddings, and how such changes in representation further influence alignment.

**Representational Convergence and the Platonic Hypothesis.** Parallel to model-brain comparison, growing evidence suggests that artificial models themselves are converging toward a universal representational structure. The systematic study of representational similarity across neural networks was pioneered by Kornblith et al. (2019), who showed that better-performing models tend to learn more similar representations. Subsequent work examined how similarity varies with width and depth (Nguyen et al., 2020). Huh et al. (2024) extended this perspective to propose the

Platonic Representation Hypothesis, which posits that sufficiently powerful learning systems naturally converge to a shared, optimal statistical model of the underlying reality. In this view, both biological and artificial intelligence are approximations of the same ideal representation.

**Intrinsic Dimension and Generalization.** Previous studies have linked representational dimension to generalization (Ansuini et al., 2019; Sharma & Kaplan, 2022). While Ansuini et al. (2019) used neighborhood-based estimates to characterize per-layer dimension, these estimates are computed with a fixed neighborhood size and do not systematically examine how the relationship between dimension and generalization varies across scales, from fine-grained local neighborhoods to more global representational structure. In contrast, we perform a multi-scale analysis and find that it is specifically the local intrinsic dimension, computed at fine spatial scales in representational space, that most sensitively tracks generalization, AI-AI alignment, and AI-Brain alignment simultaneously.

## 3. Preliminaries and Technical Background

This section introduces the datasets, notation and technical components used throughout the paper. We first describe the experimental dataset and its paired stimulus–response structure (Section 3.1). We then outline the AI–fMRI alignment pipeline and the extraction of model embeddings (Section 3.2), followed by the AI–AI alignment procedure used to compare representations across models (Section 3.3). Finally, we introduce intrinsic dimension and describe the estimator employed in our analysis (Section 3.4).

### 3.1. Dataset Description

We use the *Natural Scenes Dataset (NSD)* (Allen et al., 2022), a large-scale benchmark for studying visual representations in the human brain. The dataset includes 8 participants scanned with 7T fMRI while viewing natural scene images largely drawn from the Microsoft COCO dataset. In total, NSD contains approximately 73,000 images. Each participant viewed roughly 9,000–10,000 images, including a shared set of 1,000 images presented across all participants, resulting in approximately 22,000–30,000 trials per subject. The fMRI recordings were acquired at high spatial resolution (approximately 1.8 mm isotropic voxels), enabling fine-grained analyses of visual cortical representations.

NSD has become a widely adopted benchmark in computational neuroscience and neuro-AI research, and has been used extensively in studies of encoding and decoding models, representational similarity analysis, image reconstruction, and brain–model alignment (Doerig et al., 2023; Takagi & Nishimoto, 2023; Scotti et al., 2023; Doerig et al., 2025; Prince et al., 2024). This paired stimulus–response design

enables direct stimulus-level assessment of model–brain alignment. Given a visual stimulus, the same image can be fed into a vision model to extract internal representations, which are then compared to the neural responses elicited by that image.

### 3.2. AI–fMRI Alignment Pipeline

For a given pretrained vision model, we first compute embeddings for all images in the dataset, yielding a feature matrix in which each row corresponds to an image and each column to a feature dimension. The images are then split into training and test sets. To control for differences in embedding dimension across models, we fit a PCA on the training embeddings and project both training and test embeddings onto the top 300 principal components to avoid data leakage. For each voxel, we train a ridge regression model to predict its fMRI responses from the model embeddings using the training set. Performance is evaluated on the held-out test set using the coefficient of determination ($R^2$), which we refer to as the *alignment score*. Further details on preprocessing, cross-validation, and hyperparameter selection are provided in Appendix B.

### 3.3. AI-AI alignment pipeline

To assess alignment between different models, we adopt an embedding-based procedure that mirrors the AI–fMRI alignment pipeline. For a shared set of images, embeddings are computed for each model and the images are split into training and test sets. PCA is fit on the training embeddings and used to project both training and test embeddings onto the top 300 principal components, ensuring that dimension differences do not bias the regression. Using ridge regression, we predict one model's embeddings from another's on the training set and evaluate predictive performance on the held-out test set using $R^2$. This $R^2$ serves as the alignment score between models, providing a measure of how similarly different models represent the same input data. Since predicting Model A from Model B and Model B from Model A can yield slightly different $R^2$ values, we compute both directions and take the mean as the final alignment score. By mirroring the AI–fMRI procedure, this approach enables a consistent comparison of representations across both brain and model spaces.

### 3.4. Intrinsic dimension and its estimation

Given a set of embeddings $Z = \{z_i\}_{i=1}^{N} \subset \mathbb{R}^d$, we quantify their intrinsic dimension, which captures the complexity of the embedding set.

For an anchor point $z \in Z$, let $T_j(z)$ denote the Euclidean distance to its $j$-th nearest neighbor, and $\mathcal{N}_K(z)$ the set of its $K$ nearest neighbors. The local intrinsic dimension at $z$

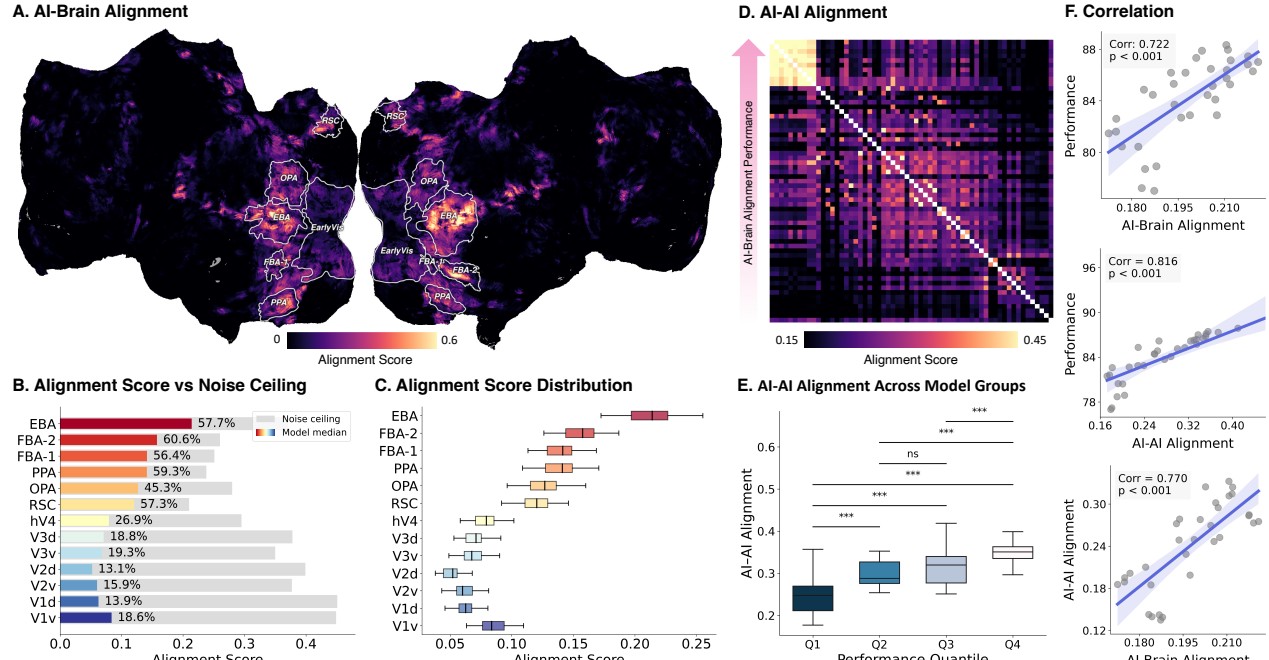

*Figure 2.* **Representational convergence links AI–Brain alignment, inter-model alignment, and generalization.** (**A**) Whole-brain maps of AI–Brain alignment, showing strongest alignment in visual cortex. (**B**) Median alignment across cortical regions, normalized by noise ceiling; higher-level visual areas (e.g., EBA) reach up to ∼60% of ceiling. (**C**) Distributions of alignment scores across models for each region, revealing substantial inter-model variability. (**D**) Pairwise AI–AI alignment matrix, with models ordered by AI–Brain alignment; models with stronger brain alignment also exhibit stronger mutual alignment. (**E**) Distributions of AI–AI alignment within groups of models binned by ImageNet performance; higher-performing groups exhibit stronger inter-model alignment. (**F**) Pairwise correlations between AI–Brain alignment, AI–AI alignment, and generalization performance, showing that these measures are all positively associated across models.

is estimated as

$$\hat{m}_K(z) = \left[ \frac{1}{K-1} \sum_{j=1}^{K-1} \log\left( \frac{T_K(z)}{T_j(z)} \right) \right]^{-1},$$

and the scale-$K$ intrinsic dimension is obtained by averaging over all points,

$$\bar{m}(K) = \frac{1}{|Z|} \sum_{z \in Z} \hat{m}_K(z).$$

The neighborhood size $K$ acts as a spatial scale: small $K$ captures local geometry, whereas larger $K$ produces a more global estimate of dimension. By varying $K$, $\bar{m}(K)$ characterizes representational complexity across scales. Additional details are provided in Appendix D.

In this work, we use the maximum-likelihood estimator (MLE) of Levina and Bickel (Levina & Bickel, 2004) to compute $\hat{m}_K(z)$. For completeness, we also compared alternative estimators, including the Method of Moments algorithm (MOM) (Amsaleg et al., 2018) and Manifold-Adaptive Dimension Estimation algorithm (MADA) (Farahmand et al., 2007), in Appendix G.

## 4. Results

### 4.1. Evidence for representational convergence

We first ask whether artificial vision models and the human visual system converge toward a shared representational structure. To address this question, we analyze representational alignment at three complementary levels: alignment between model representations and fMRI responses (AI–Brain), similarity across model representations (AI–AI), and the relationship of both to generalization performance. Our goal is to determine whether improvements in generalization are systematically accompanied by convergence in representation space.

**AI–Brain alignment across visual cortex.** We quantify alignment between pretrained vision models and human fMRI responses elicited by the same set of natural images. As shown in Figure 2A, the distribution of alignment scores across the whole brain highlights that the regions with the strongest alignment are predominantly in the visual cortex. Figure 2B summarizes the median alignment score across models for each brain region, alongside estimated noise ceilings derived from repeated fMRI measurements (for details,

see Appendix B). The noise ceiling reflects the theoretical maximum Alignment score that could be achieved. In higher-level visual areas such as EBA and FBA, the median alignment of large models reaches roughly 60% of this ceiling, indicating a high degree of correspondence. Nonetheless, Figure 2C shows substantial variability in alignment across models even within a single region such as EBA, indicating considerable differences between models.

**Representational convergence across models.** We next examine representational similarity across artificial models. Figure 2D shows the pairwise AI–AI alignment matrix, with models ordered by AI–Brain alignment performance. This ordering reveals a clear pattern: models that align more strongly with the brain also exhibit higher mutual alignment with other models, forming a coherent high-alignment block, whereas lower-performing models show weaker and more heterogeneous alignment. This observation suggests that brain-aligned models tend to converge with one another, but it remains unclear whether convergence also relates directly to generalization performance.

**Convergence increases systematically with generalization.** To test this directly, we group models according to ImageNet-1K performance and compute average within-group AI–AI alignment. As shown in Figure 2E, within-group alignment increases monotonically with generalization performance. This confirms that convergence is not specific to models selected for brain alignment, but is consistently observed among models that generalize well.

**Reference-based alignment links convergence, brain alignment, and performance.** To facilitate a quantitative analysis of AI–AI alignment, we define a reference-based measure using the highest-performing model (in terms of generalization) as a reference. Importantly, this reference is chosen solely based on performance and does not rely on neural data. By contrast, using the AI–Brain best model as a reference would produce a trivially correlated result between AI–AI and AI–Brain alignment, which we want to avoid.

As shown in Figure 2F, reference-based AI–AI alignment is strongly correlated with both AI–Brain alignment and generalization performance. Models that are closer to the generalization-optimal reference not only generalize better, but also align more strongly with neural responses, confirming that convergence in representation space reflects meaningful performance-related structure rather than trivial correlation.

Together, these analyses provide converging evidence that improved generalization is accompanied by systematic convergence in representation space. As performance increases, model representations become increasingly similar to one

another and increasingly aligned with the human visual cortex, suggesting the emergence of a shared representational structure.

### 4.2. Intrinsic dimension captures model convergence and generalization

Having established that AI representations converge with one another and with neural responses, we next ask whether a simple, measurable property of these representations can simultaneously characterize AI–AI alignment, AI–Brain alignment, and generalization performance. We focus on intrinsic dimension as a task-agnostic descriptor of representational geometry.

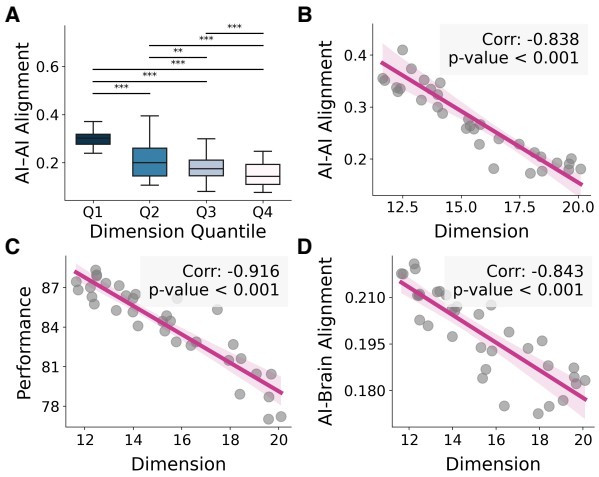

*Figure 3.* **Intrinsic dimension and representational convergence.** (**A**) Models are grouped into bins according to intrinsic dimension, and the distribution of within-group AI–AI alignment is shown for each bin; lower-dimensional models align more strongly with each other. (**B**) Intrinsic dimension is negatively correlated with AI–AI alignment measured relative to the generalization-optimal reference model. (**C**) Intrinsic dimension is negatively correlated with AI–Brain alignment in EBA. (**D**) Intrinsic dimension is negatively correlated with ImageNet-1K performance. Across all measures, lower intrinsic dimension consistently corresponds to stronger representational alignment and improved generalization.

**Dimension organizes representational convergence.** We first group models into four bins according to intrinsic dimension and examine AI–AI alignment within each group. As shown in Figure 3A, models with lower intrinsic dimension exhibit substantially stronger within-group alignment, whereas higher-dimensional models show weaker and more dispersed alignment. This demonstrates that representational convergence across models is systematically structured by intrinsic dimension.

**Dimension predicts alignment and generalization.** We next analyze intrinsic dimension at the level of individual models. Figures 3B–D show that lower intrinsic dimension

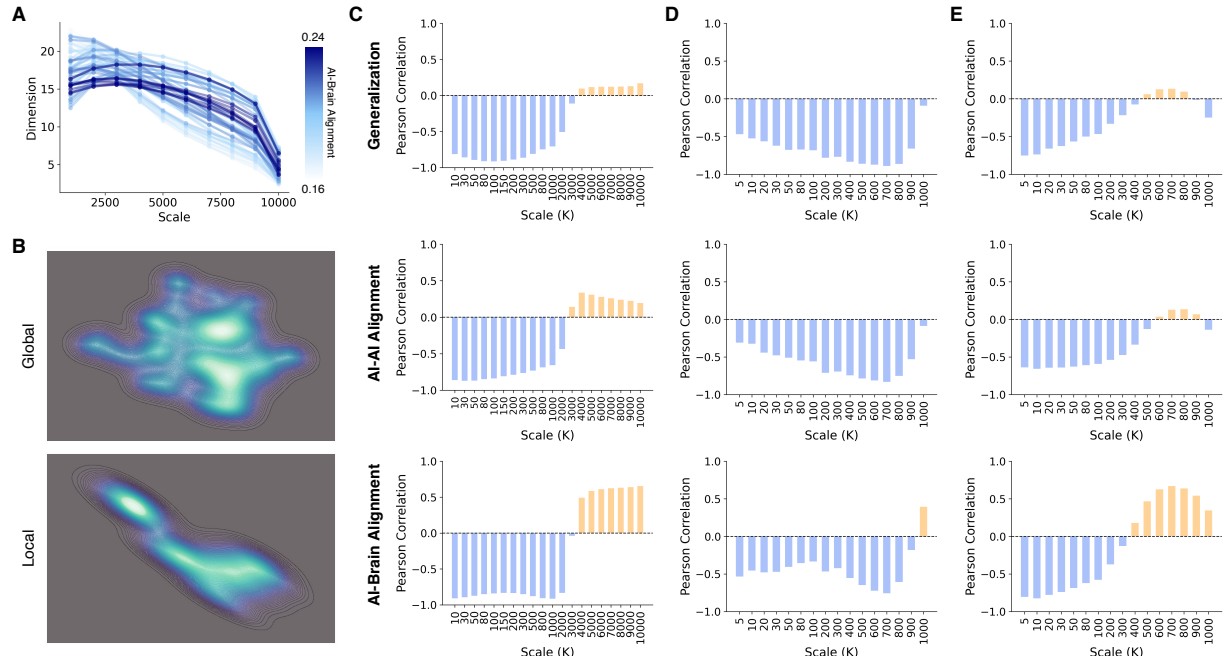

*Figure 4.* **Alignment and generalization depend on local, not global, intrinsic geometry.** (**A**) Intrinsic dimension estimated as a function of neighborhood size $K$, spanning local to global scales. (**B**) Visualization of local versus global embedding structure with matched sample sizes. (**C**) Correlations between intrinsic dimension and AI–AI alignment, AI–Brain alignment, and generalization across scales, with strongest effects at local scales. (**D**) Correlations computed within a fixed local neighborhood (1,000 nearest neighbors), remaining stable across a range of $K$. (**E**) Control analysis using random global subsampling with matched sample size, demonstrating that the observed effects are not driven by differences in data volume.

is significantly associated with stronger AI–AI alignment to the reference model, stronger AI–Brain alignment, and higher generalization performance. All relationships are statistically significant.

These results suggest that intrinsic dimension provides a simple and interpretable measure of representational convergence. Models with more compressed, lower-dimensional representations are more similar to each other, align better with neural responses, and generalize better, suggesting that geometric simplicity is associated with both biological relevance and functional performance.

### 4.3. Alignment depends on local rather than global structure

We previously analyzed intrinsic dimension at a specific scale (K=100). Here, we extend this analysis across multiple scales to ask whether the relationship between intrinsic dimension, alignment and generalization depends on the spatial scale at which dimension is estimated.

**Local dimension is most predictive.** We first estimate intrinsic dimension across a range of neighborhood sizes (Figure 4A), observing that estimated dimension systematically decreases as neighborhood size increases. To further

interpret these results, we visualized the embeddings at both global and local scales. For the global distribution, we randomly sampled 1,000 embeddings, while for the local distribution, we selected an anchor point and extracted its 1,000 nearest neighbors. Using t-SNE, we find that the global embeddings contain more distinct clusters with greater heterogeneity in inter-point distances, a hallmark of low-dimensional structure, whereas the local neighborhoods also form clusters, but fewer and more evenly distributed, consistent with higher local dimension (Figure 4B). These visualizations corroborate our multi-scale analysis and highlight the distinct geometric properties captured at different scales.

Next, in Figure 4C, we compute correlations between dimension at different scales and three measures. We find that dimension estimated at small (local) scales is most strongly predictive, whereas correlations weaken as the neighborhood size increases, indicating that alignment- and performance-relevant structure is more closely associated with local geometry.

**Local geometry, not sample size, drives the effect.** To ensure that these effects are due to local structure rather than estimation hyperparameters, we fix the neighborhood size and compute dimension within locally sampled neighbor-

hoods. Specifically, for each model, we randomly select a reference sample and extract its 1,000 nearest neighbors in embedding space to estimate local dimension. As shown in Figure 4D, the resulting dimension remains significantly negatively correlated with generalization and alignment metrics across neighborhood sizes, confirming that local geometric structure, rather than hyperparameters, governs these relationships. For comparison, we perform the same analysis on subsets obtained by randomly sampling 1,000 embeddings from the entire dataset (Figure 4E). In this case, correlations weaken and show scale-dependent effects, demonstrating that the association between dimension and performance is specifically tied to local embedding structure, not to sample count or estimation procedure.

Together, these analyses show that representational alignment and generalization are primarily associated with local, rather than global, geometry.

### 4.4. Architecture-dependent structure of representational alignment

The analyses above focused on ConvNeXt models. We next extend these observations to a broader set of architectures, ViT (3 models), a subset of ConvNeXt models (20 out of 60), ResNet (20 models), and ResMLP (8 models), to examine whether architectural differences substantially affect representational convergence, brain alignment, and generalization. To reduce the influence of hyperparameter choices, we selected the optimal estimation scale for each analysis based on the results from the previous section (AI-AI alignment and generalization used K=50, AI-Brain alignment used K=1000). Details of the models are provided in the Appendix C.

**Alignment and dimension correlations persist across architectures.** Figures 5A–F summarize relationships between AI–AI alignment (relative to the generalization-optimal reference), AI–Brain alignment, generalization performance, and intrinsic dimension across all models. As expected, combining models from different architectures slightly reduces the strength of correlations compared with within-architecture analyses. Nevertheless, the overall relationships remain statistically significant, indicating that the fundamental links between dimension, alignment, and performance are robust across heterogeneous model architectures.

**Architectural family shapes inter-model alignment.** The full pairwise AI–AI alignment matrix (Figure 5G) reveals a strong architecture-dependent pattern in inter-model alignment. Models from the same architectural family, such as ConvNeXt or ResNet, form distinct clusters with notably higher within-family alignment, whereas alignment across architectural families is substantially lower. This

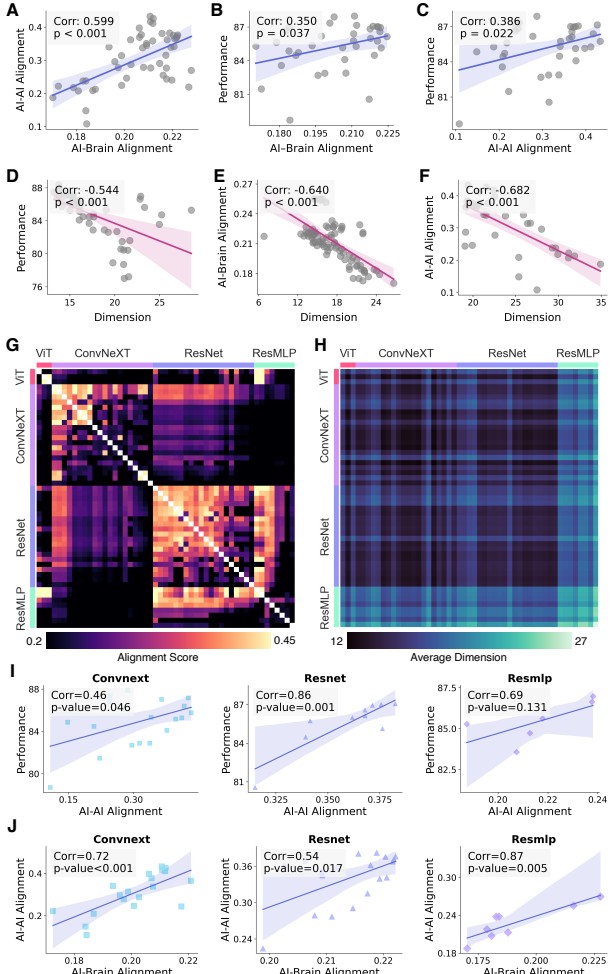

*Figure 5.* **Representational alignment generalizes across architectures.** (**A–C**) Relationships between AI–AI alignment (measured relative to the generalization-optimal reference), AI–Brain alignment, and ImageNet-1K performance when models from all architectures are pooled. (**D–F**) Relationships between intrinsic dimension and AI–AI alignment, AI–Brain alignment, and generalization across architectures. (**G**) Pairwise AI–AI alignment matrix reveals family-level structure, with stronger alignment within architectural families than across families. (**H**) Average intrinsic dimension shows no systematic differences across architectures. (**I**) Correlations between generalization performance and AI–AI alignment across models from different architectures. (**J**) Correlations between AI–AI alignment and AI–Brain alignment across architectures.

architectural clustering is not simply explained by intrinsic dimension: average intrinsic dimension does not vary systematically across families (Figure 5H). Thus, intrinsic dimension captures broad trends in generalization and alignment, but architectural family accounts for additional variation in inter-model alignment beyond dimension alone.

**Cross-architecture convergence emerges at high performance.** Interestingly, even models with very different

architectures show significant correlations between their alignment to the generalization-optimal reference and both generalization performance and AI–Brain alignment (Figure 5I–J). In our dataset, the highest-performing model, `convnext_large_mlp.clip_laion2b` `_soup_ft_in12k_in1k_384` from the ConvNeXt family, serves as this reference. Models from other architectural families, such as ResNet, exhibit alignment patterns that are significantly correlated with this top-performing model. These results indicate that, although architectural biases are associated with noticeable differences in representational geometry, models with sufficiently high performance tend to converge toward similar task-relevant representations, partially reducing architecture-dependent differences.

Together, these findings show that intrinsic dimension provides a robust, architecture-agnostic descriptor of representational simplicity, brain alignment, and generalization, whereas AI–AI alignment is strongly associated with architectural family. More detailed analyses within single architectures are presented in the Appendix H.

### 4.5. Training scale modulates local intrinsic dimension

Prior work shows that larger models and datasets improve generalization and neural alignment (Conwell et al., 2024). We asked whether these gains are accompanied by systematic changes in local representational geometry.

We first examine the relationship with model size and find that larger models (more parameters) are associated with reduced local intrinsic dimension, along with improved alignment and generalization performance (Figure 6A).

Because most ConvNeXt were pretrained on individual datasets (ImageNet-1k, ImageNet-12k, or ImageNet-22k), we can separately assess the impact of dataset size. We find that models trained on larger pretraining datasets exhibit both stronger alignment and generalization alongside lower local intrinsic dimension (Figure 6B).

Overall, larger model scale or dataset size is associated with locally lower-dimensional representations, consistent with their enhanced neural alignment and generalization. Importantly, although parameter count and local intrinsic dimension are correlated, regression analyses in Appendix L demonstrate that local intrinsic dimension accounts for significant additional variance in AI-Brain alignment, AI-AI alignment, and generalization performance beyond model size alone, indicating that representational geometry is associated with model quality in ways that are not reducible to scale.

**A. Influence of Parameters**

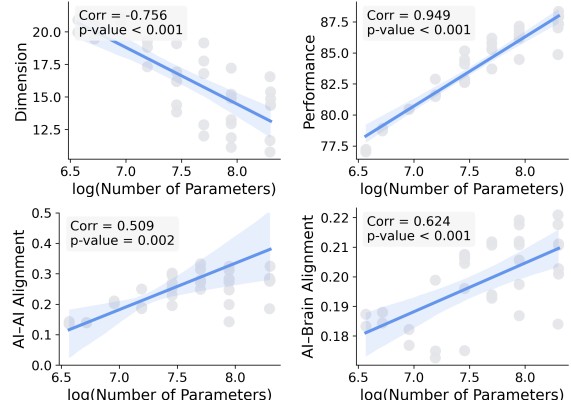

**B. Influence of Training Data Amount**

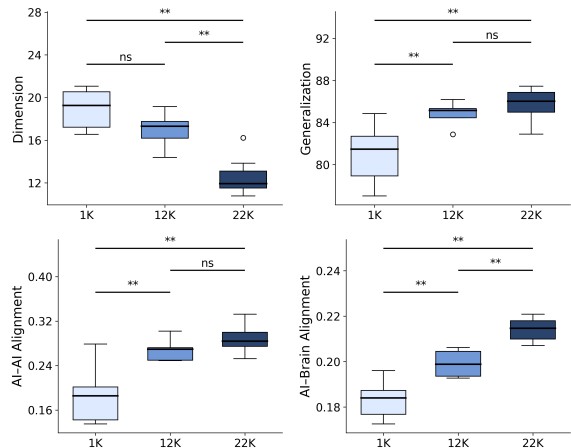

*Figure 6.* **Training scale modulates local intrinsic dimension.** (**A**) Within each architecture, larger models show lower intrinsic dimension, stronger AI–AI and AI–Brain alignment, and higher generalization. (**B**) Models trained on larger datasets exhibit decreased local dimension alongside improved alignment and performance.

## 5. Discussion

This work investigates representational convergence in modern vision models through the lens of intrinsic geometry. By jointly analyzing inter-model alignment, alignment with human neural responses, and generalization performance, we provide the first systematic evidence that higher-performing AI models exhibit increasingly similar representations both to each other and to neural representations in the brain. Crucially, these convergent representations are characterized by low local intrinsic dimension, revealing a simple geometric structure underlying alignment.

We note that all AI–Brain alignment analyses reported in the main text are robust across individual subjects and brain regions. Specifically, Appendix E presents results for different subjects, while Appendix F provides analyses across regions. These supplementary analyses confirm that the pat-

terns of alignment observed in the main text are consistent and not driven by a subset of subjects or regions.

Beyond subject and region variability, we further validate the generality of our findings along several additional aspects. Appendix I extends the analysis to language models evaluated on the MNLI benchmark, showing that the relationship among local intrinsic dimension, AI-AI alignment, and generalization is not specific to the visual domain. Appendix J demonstrates that results are robust to the choice of PCA projection dimension (80, 150, and 300 PCs). Appendix K further shows that the same overall relationships remain consistent when analyzing embeddings extracted from shallow and intermediate network layers. Taken together, these validations ensure that our conclusions are consistent and generalizable across methodological choices, modalities, and evaluation settings.

**Limitations and Future Work.** Our analyses focus on vision models and visual cortex responses. Extending this framework to other modalities will be important for assessing the generality of representational convergence. Moreover, while local intrinsic dimension strongly predicts alignment and generalization, the theoretical mechanisms underlying its relevance remain unclear.

**Conclusion.** We provide a representation-centric account of how modern vision models converge with one another and align with biological visual systems. Local intrinsic dimension emerges as a simple, architecture-agnostic link between generalization, inter-model alignment, and AI–Brain alignment. Together, these results suggest that learning and scale promote locally simple representational structures that support both model-to-model convergence and alignment with the brain.

## Acknowledgements

This work was supported by Brain Science and Brain-like Intelligence Technology - National Science and Technology Major Project (2021ZD0200500), the National Natural Science Foundation of China (62472206, 3254100307), National Key R&D Program of China (2025YFC3410000), Shenzhen Science and Technology Innovation Committee (RCYX20231211090405003, JCYJ20220818100213029), Guangdong Basic and Applied Basic Research Foundation (2026B1515020099), Guangdong S&T Program (Grant No. 2026B0101110003), Shanghai Municipal Special Program for Basic Research on General AI Foundation Models (2025SHZDZX026D05), GuangDong Basic and Applied Basic Research Foundation (2025A1515011645), GuangDong Basic and Applied Basic Research Foundation (2026A1515010121), Shenzhen Doctoral Startup Project (RCBS20231211090748082), Shenzhen Loop Area Institute under grant FPF10120250012, and the open research fund of the Guangdong Provincial Key Laboratory of Mathematical and Neural Dynamical Systems, the Center for Computational Science and Engineering at Southern University of Science and Technology, Shenzhen Key Laboratory of Smart Healthcare Engineering.

## Impact Statement

This paper presents work whose goal is to advance the field of machine learning. Our study investigates representational alignment between AI models and biological visual systems, with potential implications for model interpretability and generalization. This work is analytical and does not introduce a deployed system or new neural data-collection protocol. There are many potential societal consequences of this work, none of which we feel must be specifically highlighted here.

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

# A. Overview of Supplementary Analyses

This appendix provides a comprehensive set of analyses that extend and validate the main findings of the study. While the main text focuses on EBA and ConvNeXt embeddings, the supplementary analyses explore multiple aspects to ensure the robustness and generality of our conclusions:

- **fMRI Alignment Details (Section B)**: Detailed description of voxel-wise encoding models, evaluation metrics, and noise ceiling estimation for transparency and reproducibility.

- **Model Architectures and Implementation (Section C)**: Examination of multiple architectures (ConvNeXt, ResNet, ResMLP, ViT) with various pretraining regimes to test cross-model generality.

- **Dimensional Analysis (Section D)**: Methods for estimating intrinsic dimension, including fractal and correlation dimensions, with multi-scale analysis to capture geometric structure of model representations.

- **Subject Differences (Section E)**: Assessment of inter-subject variability in AI-Brain alignment, its correlation with generalization, and dimension to ensure findings are not driven by individual participants.

- **Region Differences (Section F)**: Evaluation across early, intermediate, and higher-level visual areas to verify region-general and hierarchical patterns in AI-Brain alignment.

- **Dimension Estimator Robustness (Section G)**: Comparison of multiple estimators (MLE, MOM, MADA) to confirm scale-dependent dimension effects are robust to methodological choices.

- **Supplementary Analysis of Architecture Differences (Section H)**: Breakdown of alignment and dimension results by architecture family, verifying that the observed relationships hold within and across distinct architectural groups rather than being driven by architectural confounds.

- **Results of Language Models (Section I)**: Extension of the main analysis to language models on the MNLI task, demonstrating that the relationship between AI-AI alignment, generalization, and local intrinsic dimension generalizes beyond vision models.

- **Validation of PC Dimension (Section J)**: Robustness check of the PCA projection dimension used in the main analysis, repeating all experiments with 80 and 150 principal components to verify that conclusions are not sensitive to this choice.

- **Details of Layer-wise Analysis (Section K)**: Analysis of embeddings extracted from intermediate layers (10%, 30%, 50%, and 100% depth), showing how local intrinsic dimension evolves across network depth and confirming that the main findings hold at early and intermediate layers.

- **Incremental Predictive Power of Local Intrinsic Dimension (Section L)**: Multiple regression analyses demonstrating that local intrinsic dimension explains significant additional variance in AI-Brain alignment, AI-AI alignment, and generalization performance beyond what is accounted for by model parameter count alone.

Collectively, these analyses validate the reproducibility and generality of our findings, demonstrating that local intrinsic dimension reliably predicts representational alignment and generalization across model types, evaluation benchmarks, projection choices, brain regions, participants, and network depths.

# B. Details of fMRI Alignment

**Voxel-wise encoding model.** Let $z_i \in \mathbb{R}^{300}$ be the embedding of image $i$, and let $r_{ij}$ be the fMRI response of voxel $j$ to image $i$. We train a voxel-specific ridge regression model:

$$r_{ij} \approx \mathbf{w}_j^\top \mathbf{z}_i + \epsilon. \tag{1}$$

Model fitting is done using 5-fold cross-validation on an 80/20 training/test split. The regularization parameter is tuned via nested cross-validation.

**Evaluation metrics.**   Alignment quality is quantified by the coefficient of determination ($R^2$) on held-out data.

**Noise ceiling estimation.**   Due to inherent measurement errors and individual variability in neural signals, even a perfectly accurate model cannot account for all sources of variation. Noise ceiling estimates are derived from the reliability of the beta weights across trials. In essence, the more consistent the neural response is across repeated presentations of the same image, the greater the proportion of the response variance that can be attributed to stimulus-driven signals. These estimates of the noise ceiling establish an upper bound on the amount of variance that can be explained or predicted in the response of a given voxel (Allen et al., 2022).

To quantify the noise ceiling in a more precise manner, the effective noise variance $N_{\text{eff}}$ is first computed by considering the number of times each image was presented to the subject, with different weights assigned to each trial type. Let $A$, $B$, and $C$ denote the number of distinct images presented three, two, and one times to the subject, respectively. The effective noise variance $N_{\text{eff}}$ is calculated as:

$$N_{\text{eff}} = \frac{\frac{A}{3} + \frac{B}{2} + \frac{C}{1}}{A + B + C}$$

Subsequently, the noise ceiling $N_C$ is determined by combining the signal variance $S^2$ with the effective noise variance $N_{\text{eff}}$, as follows:

$$N_C = \frac{S^2}{S^2 + N_{\text{eff}}}$$

Here, $S^2$ represents the signal variance, which is computed based on the beta weights derived from all NSD scan sessions. The data used is provided by the NSD dataset.

## C. Model Architectures and Implementation Details

In this study, we analyzed multiple convolutional and transformer-based architectures to examine the alignment between neural representations and model embeddings. Our analysis primarily focuses on the ConvNeXt family, while ResNet, ResMLP, and ViT models serve as cross-architecture controls to validate the generality of our findings.

### C.1. ConvNeXt: Primary Analysis Backbone

We employed the ConvNeXt architecture (Liu et al., 2022) as the main backbone for generating embeddings. ConvNeXt adapts the ResNet architecture with design principles inspired by Vision Transformers, including large kernel sizes, depthwise convolutions, and layer normalization. Its architecture consists of four stages with multiple residual blocks per stage. Key modifications include:

- **GELU activation**: replaces ReLU for smoother gradient propagation.

- **Large convolutional kernels (e.g., 7x7)**: increases the receptive field.

- **Depthwise separable convolutions**: efficient spatial mixing while reducing parameters.

- **Layer normalization instead of batch normalization**: stabilizes training across different datasets.

- **Patchify stem**: aligns preprocessing with Vision Transformer-style models.

**Advantages of ConvNeXt for controlled analysis:**

1. **Controlled pretraining**: Each model is pre-trained on a single dataset, allowing systematic comparisons of pretraining effects on brain-model alignment.

2. **Architectural similarity**: All models share the same backbone, minimizing confounding effects from structural differences.

*Table 1.* Vision-only ConvNeXt models without language supervision.

| Model Name | Pretraining Dataset | FT |
|---|---|---|
| convnext_nano.d1h_in1k | ImageNet-1K | ✗ |
| convnext_nano_ols.d1h_in1k | ImageNet-1K | ✗ |
| convnext_tiny_hnf.a2h_in1k | ImageNet-1K | ✗ |
| convnext_pico_ols.d1_in1k | ImageNet-1K | ✗ |
| convnext_pico.d1_in1k | ImageNet-1K | ✗ |
| convnext_atto_ols.a2_in1k | ImageNet-1K | ✗ |
| convnext_large.fb_in1k | ImageNet-1K | ✗ |
| convnext_femto.d1_in1k | ImageNet-1K | ✗ |
| convnext_base.fb_in1k | ImageNet-1K | ✗ |
| convnext_atto.d2_in1k | ImageNet-1K | ✗ |
| convnext_femto_ols.d1_in1k | ImageNet-1K | ✗ |
| convnext_small.in12k_ft_in1k_384 | ImageNet-12K | ✓ |
| convnext_small.in12k_ft_in1k | ImageNet-12K | ✓ |
| convnext_small.fb_in1k | ImageNet-1K | ✗ |
| convnext_tiny.fb_in1k | ImageNet-1K | ✗ |
| convnext_nano.in12k_ft_in1k | ImageNet-12K | ✓ |
| convnext_tiny.in12k_ft_in1k | ImageNet-12K | ✓ |
| convnext_tiny.in12k_ft_in1k_384 | ImageNet-12K | ✓ |
| convnext_tiny.fb_in22k_ft_in1k_384 | ImageNet-22K | ✓ |
| convnext_tiny.fb_in22k_ft_in1k | ImageNet-22K | ✓ |
| convnext_small.fb_in22k_ft_in1k_384 | ImageNet-22K | ✓ |
| convnext_small.fb_in22k_ft_in1k | ImageNet-22K | ✓ |
| convnext_small.in12k | ImageNet-12K | ✗ |
| convnext_base.fb_in22k_ft_in1k_384 | ImageNet-22K | ✓ |
| convnext_large.fb_in22k_ft_in1k_384 | ImageNet-22K | ✓ |
| convnext_base.fb_in22k_ft_in1k | ImageNet-22K | ✓ |
| convnext_xlarge.fb_in22k_ft_in1k_384 | ImageNet-22K | ✓ |
| convnext_tiny.in12k | ImageNet-12K | ✗ |
| convnext_large.fb_in22k_ft_in1k | ImageNet-22K | ✓ |
| convnext_xlarge.fb_in22k_ft_in1k | ImageNet-22K | ✓ |
| convnext_nano.in12k | ImageNet-12K | ✗ |
| convnext_small.fb_in22k | ImageNet-22K | ✗ |
| convnext_large.fb_in22k | ImageNet-22K | ✗ |
| convnext_tiny.fb_in22k | ImageNet-22K | ✗ |
| convnext_xlarge.fb_in22k | ImageNet-22K | ✗ |
| convnext_base.fb_in22k | ImageNet-22K | ✗ |

3. **Scalability across model sizes**: ConvNeXt variants range from `nano` to `xlarge`, enabling analyses across scales while keeping architectural principles consistent.

4. **Computational efficiency**: Its design allows us to extract embeddings for multiple participants and brain regions without excessive hardware requirements.

**Implementation Details:** We utilized the Huggingface Transformers API to access ConvNeXt models pre-trained on ImageNet-1K, ImageNet-12K, ImageNet-22K, and LAION datasets. Embeddings were extracted on an NVIDIA RTX 3080 GPU with 10GB memory, sufficient for all model variants. Table 1 and Table 2 list all ConvNeXt models, including pretraining datasets and whether fine-tuning was applied.

**Computational Resources** We utilized an NVIDIA GeForce RTX 3080 GPU with 10GB GDDR6X memory, a 320-bit memory interface, and a memory bandwidth of 760 GB/s for embedding extraction using the Huggingface Transformers API. This setup adequately meets the computational requirements without requiring additional resources.

*Table 2.* Vision–language ConvNeXt models with language supervision.

| Model Name | Pretraining Dataset | FT |
|---|---|---|
| convnext_base.clip_laion2b_augreg_ft_in1k | LAION-2B | ✓ |
| convnext_large_mlp.clip_laion2b_augreg_ft_in1k | LAION-2B | ✓ |
| convnext_large_mlp.clip_laion2b_augreg_ft_in1k_384 | LAION-2B | ✓ |
| convnext_base.clip_laiona_augreg_ft_in1k_384 | LAION-A | ✓ |
| convnext_large_mlp.clip_laion2b_soup_ft_in12k_in1k_384 | LAION-2B | ✓ |
| convnext_base.clip_laion2b_augreg_ft_in12k_in1k | LAION-2B | ✓ |
| convnext_large_mlp.clip_laion2b_soup_ft_in12k_in1k_320 | LAION-2B | ✓ |
| convnext_base.clip_laion2b_augreg_ft_in12k_in1k_384 | LAION-2B | ✓ |
| convnext_xxlarge.clip_laion2b_soup_ft_in1k | LAION-2B | ✓ |
| convnext_large_mlp.clip_laion2b_soup_ft_in12k_384 | LAION-2B | ✓ |
| convnext_large_mlp.clip_laion2b_augreg_ft_in12k_384 | LAION-2B | ✓ |
| convnext_base.clip_laion2b_augreg_ft_in12k | LAION-2B | ✓ |
| convnext_xxlarge.clip_laion2b_soup_ft_in12k | LAION-2B | ✓ |
| convnext_large_mlp.clip_laion2b_soup_ft_in12k_320 | LAION-2B | ✓ |
| convnext_xxlarge.clip_laion2b_rewind | LAION-2B | ✗ |
| convnext_xxlarge.clip_laion2b_soup | LAION-2B | ✗ |
| convnext_large_mlp.clip_laion2b_augreg | LAION-2B | ✗ |
| convnext_large_mlp.clip_laion2b_ft_320 | LAION-2B | ✓ |
| convnext_base.clip_laiona | LAION-A | ✗ |
| convnext_large_mlp.clip_laion2b_ft_soup_320 | LAION-2B | ✓ |
| convnext_base.clip_laion2b | LAION-2B | ✗ |
| convnext_base.clip_laiona_320 | LAION-A | ✗ |
| convnext_base.clip_laion2b_augreg | LAION-2B | ✗ |
| convnext_base.clip_laiona_augreg_320 | LAION-A | ✗ |

## C.2. Cross-Architecture Validation Models

To ensure that our findings from ConvNeXt are not architecture-specific, we also evaluated several additional families of models: ResNet, ResMLP, and Vision Transformers (ViT). These models allow us to test whether trends observed in ConvNeXt embeddings generalize across architectural paradigms.

### C.2.1. RESNET MODELS

We examined ResNet models across different depths (18, 50, 101, 152 layers) and training paradigms (e.g., ImageNet-1K, supervised, self-supervised). The residual structure and hierarchical feature extraction of ResNets provide a natural comparison to ConvNeXt. Table 3 summarizes brain-region alignment results for selected ResNet variants.

### C.2.2. RESMLP MODELS

ResMLP is a pure MLP-based vision architecture that replaces convolutions with linear projections while retaining spatial information. We evaluated ResMLP models of varying depths (12, 24, 36 layers) and training regimes. Table 4 presents the alignment scores for EBA, OPA, and PPA, demonstrating that the general relationship between pretraining and brain alignment extends to MLP-based architectures.

### C.2.3. VISION TRANSFORMERS (VIT)

ViT models use self-attention to capture long-range dependencies in images. We analyzed MAE-pretrained ViT models of base, large, and huge sizes. Table 5 shows brain-region alignment scores.

## C.3. Summary

- **ConvNeXt**: Serves as the primary architecture for controlled analyses, due to consistent backbone, scalable sizes, and pretraining on single datasets.

- **ResNet, ResMLP, ViT**: Used as cross-architecture checks to verify that key trends are not specific to ConvNeXt.

*Table 3.* Summary of ResNet models across depth and training variants.

| Model | $R^2_{\text{EBA}}$ | $R^2_{\text{OPA}}$ | $R^2_{\text{PPA}}$ |
|---|---|---|---|
| resnet18.a1_in1k | 0.211 | 0.131 | 0.150 |
| resnet18.a2_in1k | 0.208 | 0.126 | 0.148 |
| resnet18.a3_in1k | 0.199 | 0.122 | 0.142 |
| resnet18.fb_ssl_yfcc100m_ft_in1k | 0.215 | 0.133 | 0.152 |
| resnet18.fb_swsl_ig1b_ft_in1k | 0.217 | 0.134 | 0.153 |
| resnet50.a1_in1k | 0.222 | 0.135 | 0.154 |
| resnet50.a1h_in1k | 0.209 | 0.126 | 0.144 |
| resnet50.a2_in1k | 0.225 | 0.140 | 0.159 |
| resnet50.a3_in1k | 0.219 | 0.138 | 0.156 |
| resnet50.am_in1k | 0.222 | 0.142 | 0.160 |
| resnet101.a1_in1k | 0.219 | 0.134 | 0.150 |
| resnet101.a1h_in1k | 0.206 | 0.121 | 0.140 |
| resnet101.a2_in1k | 0.222 | 0.137 | 0.155 |
| resnet101.a3_in1k | 0.221 | 0.136 | 0.154 |
| resnet101.gluon_in1k | 0.214 | 0.130 | 0.148 |
| resnet152.a1_in1k | 0.216 | 0.130 | 0.149 |
| resnet152.a1h_in1k | 0.203 | 0.118 | 0.135 |
| resnet152.a2_in1k | 0.220 | 0.136 | 0.152 |
| resnet152.a3_in1k | 0.218 | 0.136 | 0.154 |
| resnet152.gluon_in1k | 0.209 | 0.126 | 0.144 |

*Table 4.* Summary of ResMLP models.

| Model | $R^2_{\text{EBA}}$ | $R^2_{\text{OPA}}$ | $R^2_{\text{PPA}}$ |
|---|---|---|---|
| resmlp_12_224.fb_dino | 0.216 | 0.144 | 0.158 |
| resmlp_12_224.fb_distilled_in1k | 0.188 | 0.111 | 0.125 |
| resmlp_12_224.fb_in1k | 0.181 | 0.107 | 0.122 |
| resmlp_24_224.fb_dino | 0.228 | 0.148 | 0.161 |
| resmlp_24_224.fb_distilled_in1k | 0.183 | 0.109 | 0.123 |
| resmlp_24_224.fb_in1k | 0.170 | 0.096 | 0.108 |
| resmlp_36_224.fb_distilled_in1k | 0.184 | 0.106 | 0.121 |
| resmlp_36_224.fb_in1k | 0.179 | 0.104 | 0.116 |

*Table 5.* Summary of ViT models pretrained with MAE.

| Model | $R^2_{\text{EBA}}$ | $R^2_{\text{OPA}}$ | $R^2_{\text{PPA}}$ |
|---|---|---|---|
| vit_base_patch16_224.mae | 0.217 | 0.151 | 0.163 |
| vit_large_patch16_224.mae | 0.227 | 0.155 | 0.165 |
| vit_huge_patch14_224.mae | 0.220 | 0.154 | 0.164 |

- **Overall**: Across all architectures, the alignment between model embeddings and neural data demonstrates systematic dependence on pretraining data and model scale, supporting the generality of our findings.

## D. Background on Dimensional Analysis

### D.1. Fractal Dimension and Correlation Dimension

The concept of fractal dimension provides a robust framework for quantifying the complexity, irregularity and structural intricacies of datasets across different scales. Unlike traditional Euclidean dimensions, fractal dimensions capture the degree to which a set fills space as the observation scale varies. This is particularly useful in the context of high-dimensional data and manifold learning.

**Definition D.1. Fractal Dimension**: The fractal dimension $D_f$ is defined as the scaling exponent that quantifies how the number of self-similar structures in a set changes with the scale of observation. Mathematically, it can be expressed as:

$$D_f = \lim_{\epsilon \to 0} \frac{\log N(\epsilon)}{\log(1/\epsilon)}, \tag{2}$$

where $N(\epsilon)$ represents the number of $\epsilon$-sized boxes required to cover the set.

A higher fractal dimension indicates a more intricate structure with greater self-similarity across multiple scales.

### D.1.1. CORRELATION DIMENSION

The correlation dimension is a specific type of fractal dimension that focuses on the spatial distribution properties of a point set, effectively capturing statistical sparsity and clustering behavior. It provides a finer-grained analysis of the dataset structure, especially in cases where data points are non-uniformly distributed.

**Definition D.2. Correlation Dimension**: The correlation dimension $D_C$ is defined based on the correlation function $C(\epsilon)$ as:

$$D_C = \lim_{\epsilon \to 0} \frac{\log C(\epsilon)}{\log \epsilon}, \tag{3}$$

where the correlation function $C(\epsilon)$ is given by:

$$C(\epsilon) = \frac{1}{N(N-1)} \sum_{i=1}^{N} \sum_{j \neq i} \mathbb{I}(\|x_i - x_j\| < \epsilon). \tag{4}$$

- $N$ is the number of data points.

- $\|x_i - x_j\|$ represents the Euclidean distance between points $x_i$ and $x_j$.

- $\mathbb{I}(\cdot)$ is the indicator function, equal to $1$ if the condition inside is true and $0$ otherwise.

The correlation dimension effectively estimates the likelihood that two randomly chosen points from the dataset are within a distance $\epsilon$ of each other, thus providing insights into the spatial organization of the data.

### D.2. Fractal Dimension Estimation Using Maximum Likelihood

To accurately estimate the fractal dimension, particularly the correlation dimension, we employ the Maximum Likelihood Estimation (MLE) method as proposed by Levina and Bickel (Levina & Bickel, 2004). This method leverages distances between neighboring points to infer the intrinsic dimension of the underlying manifold.

### D.2.1. FRACTAL DIMENSION ESTIMATION VIA MLE

The MLE-based estimation method utilizes the $k$-nearest neighbor distances to quantify the dimension of a dataset. For a given data point $x_i$, the estimated fractal dimension $\hat{m}_k(x_i)$ at a specific $k$ value is defined as:

$$\hat{m}_k(x_i) = \left[ \frac{1}{k-1} \sum_{j=1}^{k-1} \log \frac{T_k(x_i)}{T_j(x_i)} \right]^{-1}, \tag{5}$$

where:

- $T_j(x_i)$ is the Euclidean distance from $x_i$ to its $j^{\text{th}}$ nearest neighbor.

- $k$ is the number of nearest neighbors considered for the estimation.

A smaller $k$ captures local structural variations, emphasizing finer-scale features, while a larger $k$ reflects broader, more global patterns in the data distribution.

### D.2.2. AVERAGING OVER DATA POINTS

To obtain a robust estimate of the fractal dimension for the entire dataset, we average the estimated dimensions across all data points:

$$\bar{m}_k = \frac{1}{N} \sum_{i=1}^{N} \hat{m}_k(x_i), \tag{6}$$

where $N$ is the total number of data points. This averaging process helps to mitigate the effects of noise and local variations, producing a stable estimate of the fractal dimension at a specific $k$ value.

### D.2.3. SELECTION OF $k$ AND SCALE ANALYSIS

The choice of $k$ significantly impacts the resulting dimension estimate:

- **Small** $k$: Focuses on finer-scale structures, capturing local density variations and small-scale clustering.

- **Large** $k$: Emphasizes broader trends, providing a more smoothed, global perspective of the dataset structure.

To achieve a comprehensive understanding of the manifold, it is essential to analyze the fractal dimension estimates across multiple $k$ values. This multi-scale analysis helps in identifying how the estimated dimension varies with scale, offering deeper insights into the intrinsic complexity of the dataset.

## E. Supplementary Analysis of Subject Differences

### E.1. Impact of Subject Differences on AI-Brain Alignment

To further investigate the voxel-wise alignment analysis presented in the main text, we extended the analysis across all eight participants using multiple models. Here, we focus on two key visualizations that highlight the variability and consistency of alignment performance.

Figure 7 illustrates the alignment performance of a selected model across the eight participants. Despite inter-subject variability, the overall spatial distribution of alignment remains relatively consistent across participants. Regions around the EBA consistently show higher alignment, indicating that model-derived representations align more effectively with neural activity in higher-order visual areas. This consistent spatial distribution suggests that the alignment patterns observed in the main analysis are not specific to a single participant but are generalizable across multiple subjects.

### E.2. Impact of Subject Differences on the Correlation between AI-Brain and AI-AI Alignment

We examine whether the correlation between AI-Brain alignment and AI-AI alignment is consistent across individual subjects. For this analysis, we focus on the EBA region and compute, for each subject, the correlation between AI-Brain alignment and AI-AI alignment across models.

As shown in Figure 8, significant positive correlations are observed for all subjects. The results indicate that models exhibiting stronger alignment with other AI models also tend to better match brain representations, and this relationship is robust across individual differences in neural responses.

### E.3. Impact of Subject Differences on the Correlation between AI-Brain Alignment and Generalization

We further investigate whether the relationship between AI-Brain alignment and model generalization performance is affected by subject variability. For each subject in EBA, we compute the correlation between AI-Brain alignment and test performance across models.

Figure 9 shows that all subjects exhibit a significant positive correlation, indicating that models which align better with brain activity in EBA also tend to generalize more effectively. This finding confirms that the predictive power of AI-Brain alignment for generalization is consistent across individual subjects.

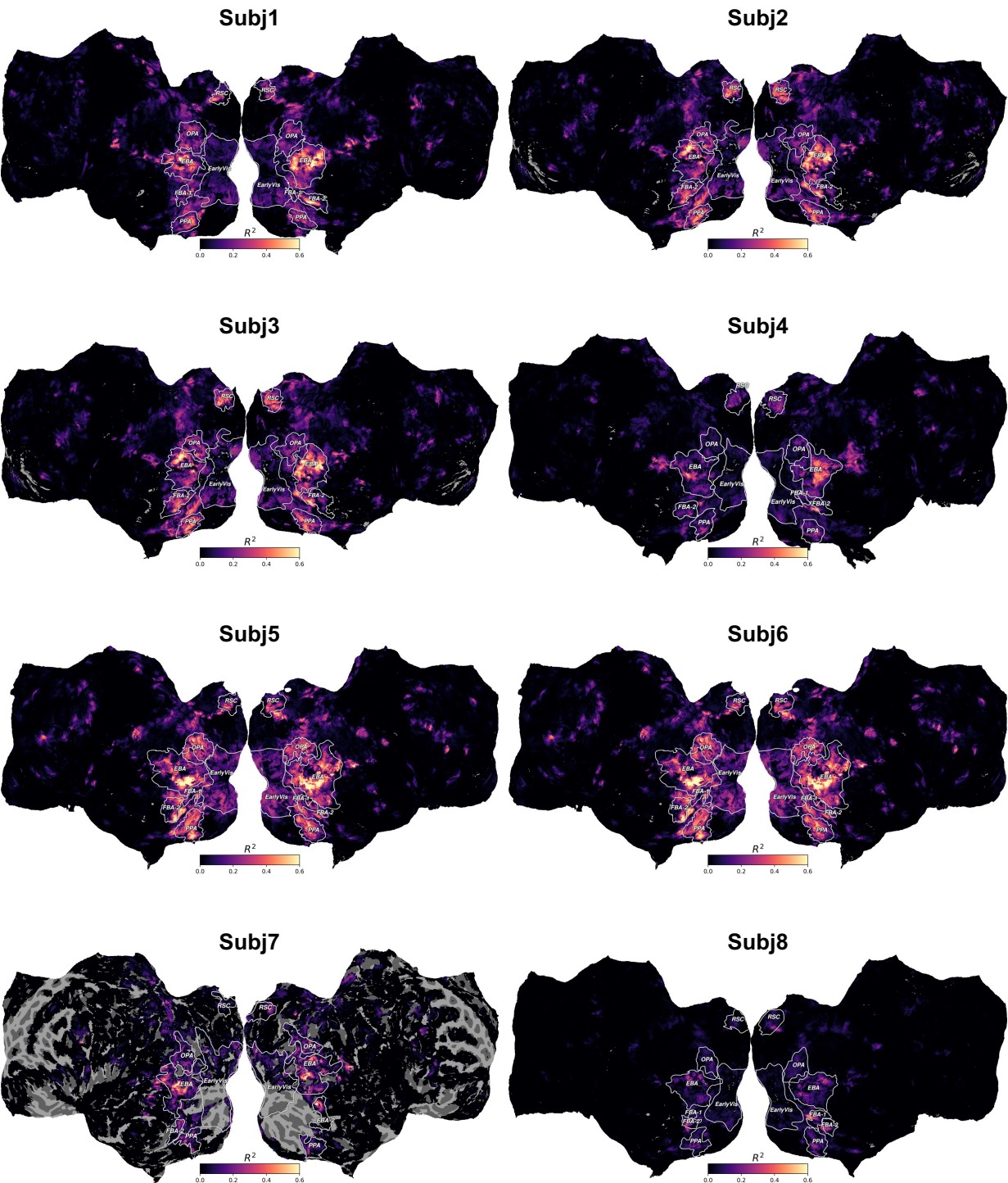

*Figure 7.* **Alignment performance across different subjects for the same model (Model index: M50).** This figure illustrates the alignment performance of the same model across different subjects. There is a noticeable variation in alignment effectiveness, with Subjects 5 and 6 exhibiting better alignment, while Subjects 7 and 8 show relatively weaker alignment. This highlights substantial inter-subject variability in alignment performance.

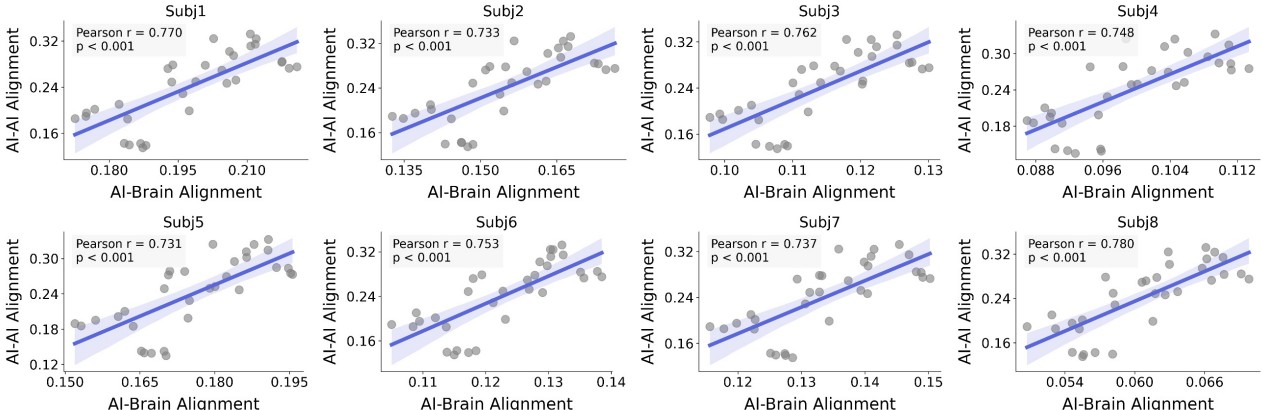

*Figure 8.* **Correlation between AI-Brain alignment and AI-AI alignment across subjects in EBA.** Each panel corresponds to a distinct subject. For each subject, we report the Pearson correlation across models. Significant positive correlations are consistently observed, demonstrating that the relationship between AI-AI and AI-Brain alignment is robust to individual differences.

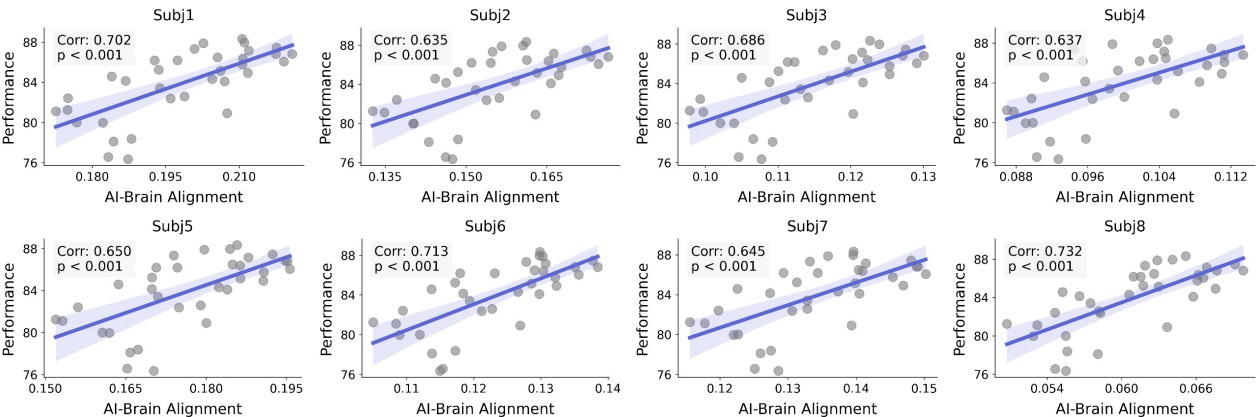

*Figure 9.* **Correlation between AI-Brain alignment and generalization performance across subjects in EBA.** Each panel corresponds to a distinct subject. For all subjects, the correlation is positive and statistically significant, indicating that alignment with EBA representations reliably predicts model generalization, independent of individual differences.

### E.4. Impact of Subject Differences on the Correlation between Dimension and AI-Brain Alignment

Finally, we assess whether the relationship between representational dimension and AI-Brain alignment varies across subjects. For each subject, we compute the correlation between the estimated intrinsic dimension of model representations and AI-Brain alignment in EBA.

As depicted in Figure 10, all subjects show significant positive correlations, suggesting that the link between dimension and brain alignment is robust to individual variability and reflects a general geometric property of model representations.

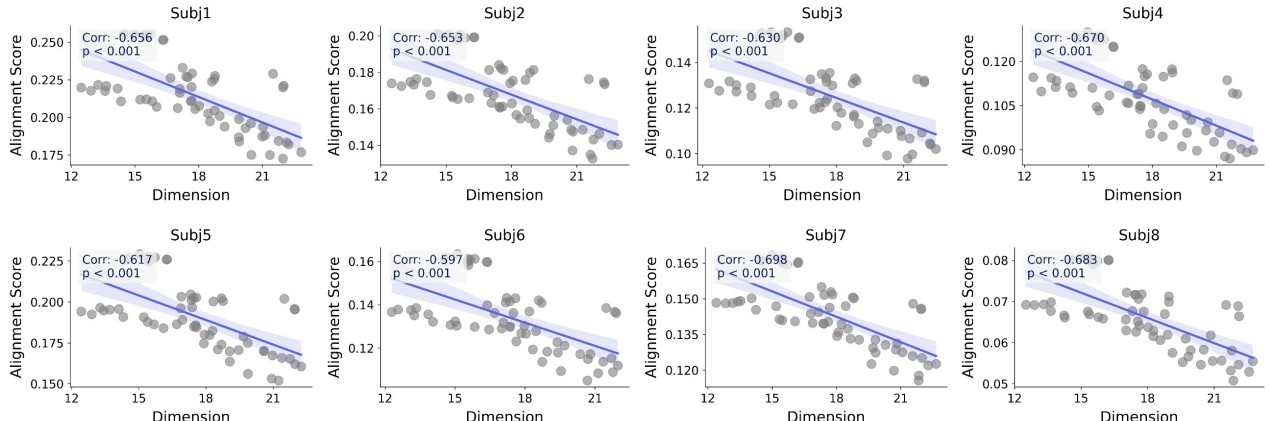

*Figure 10.* **Correlation between representational dimension and AI-Brain alignment across subjects in EBA.** Each panel corresponds to a distinct subject. Significant positive correlations are observed for all subjects, indicating that the relationship between intrinsic dimension and brain alignment is robust to individual differences.

# F. Supplementary Analysis of Region Differences

In the main text, our analysis primarily focuses on the extrastriate body area (EBA), where AI-Brain alignment shows the strongest and most stable effects. In this section, we extend our analysis to multiple visual regions of interest (ROIs) to examine whether the observed relationships generalize across the visual hierarchy and to assess the influence of regional differences on AI-Brain alignment.

### F.1. Impact of Region Differences on the Correlation between AI-Brain and AI-AI Alignment

We first examine whether the correlation between AI-Brain alignment and AI-AI alignment, as reported in the main text, is specific to EBA or persists across other visual regions. For each ROI, we compute the correlation between AI-Brain alignment and AI-AI alignment across models.

As shown in Figure 11, a significant positive correlation is consistently observed across a wide range of visual regions, including early (V1, V2), intermediate (V3, hV4), and higher-level visual areas. Although the absolute strength of the correlation varies across regions, the overall trend remains stable, indicating that models that better align with other AI models also tend to exhibit stronger alignment with brain representations, independent of the specific cortical region considered. This result demonstrates that the coupling between AI-AI and AI-Brain alignment is not restricted to a single functional area but reflects a more general representational property.

### F.2. Impact of Region Differences on the Correlation between AI-Brain Alignment and Generalization

Next, we analyze how AI-Brain alignment in different visual regions relates to model generalization performance. For each ROI, we compute the correlation between AI-Brain alignment and test performance across models.

Figure 12 reveals a clear regional dependence along the visual hierarchy. In early visual cortex (e.g., V1), the correlation between AI-Brain alignment and generalization performance is weak and not statistically significant. However, as visual information progresses through the hierarchy from V1 to V2 and hV4, the correlation strength gradually increases. In higher-level visual regions, the correlation becomes both statistically significant and substantially stronger.

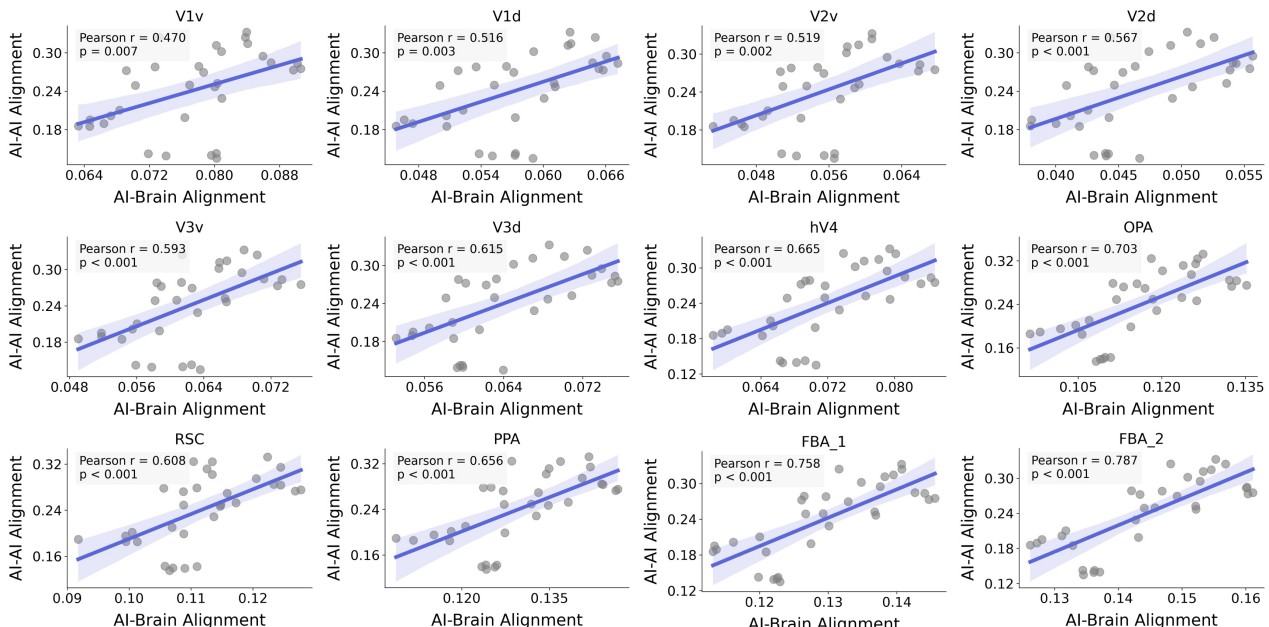

*Figure 11.* **Correlation between AI-Brain alignment and AI-AI alignment across different visual regions.** Each panel corresponds to a distinct region of interest. For each region, we report the Pearson correlation between AI-Brain alignment and AI-AI alignment across models. Significant positive correlations are consistently observed across regions, indicating that the relationship between AI-Brain and AI-AI alignment is robust to regional differences.

These results suggest that alignment with higher-order visual representations is more predictive of a model's ability to generalize, whereas alignment with early sensory representations alone is insufficient. This hierarchical pattern is consistent with the functional roles of these regions and supports the interpretation that AI-Brain alignment captures behaviorally relevant representational structure more effectively in higher visual areas.

### F.3. Impact of Region Differences on the Correlation between Dimension and AI-Brain Alignment

We further investigate whether the relationship between representational dimension and AI-Brain alignment depends on the cortical region considered. For each ROI, we compute the correlation between the estimated intrinsic dimension of model representations and AI-Brain alignment.

As shown in Figure 13, dimension is significantly correlated with AI-Brain alignment across all examined regions. Unlike the analysis with generalization performance, we do not observe a systematic increase in correlation strength from early to higher-level visual areas. Instead, the dimension-alignment relationship appears relatively stable across regions.

This result suggests that the link between representational dimension and AI-Brain alignment reflects a general geometric property of representations rather than a region-specific effect tied to higher-level visual processing.

## G. Supplementary Analysis of Dimension Estimators

In the main text, we emphasize the importance of local representational structure by analyzing dimension at different neighborhood scales. Here, we further assess the robustness of these findings with respect to the choice of dimension estimator.

### G.1. Impact of Estimator Differences on Scale Analysis

We first evaluate whether the scale-dependent relationships reported in the main text depend on a specific dimension estimation method. In addition to the estimator used in the main analysis, we consider two alternative estimators, MOM and MADA. For each estimator and neighborhood scale, we compute the correlation between estimated dimension and AI-AI alignment, AI-Brain alignment, and generalization performance using all samples.

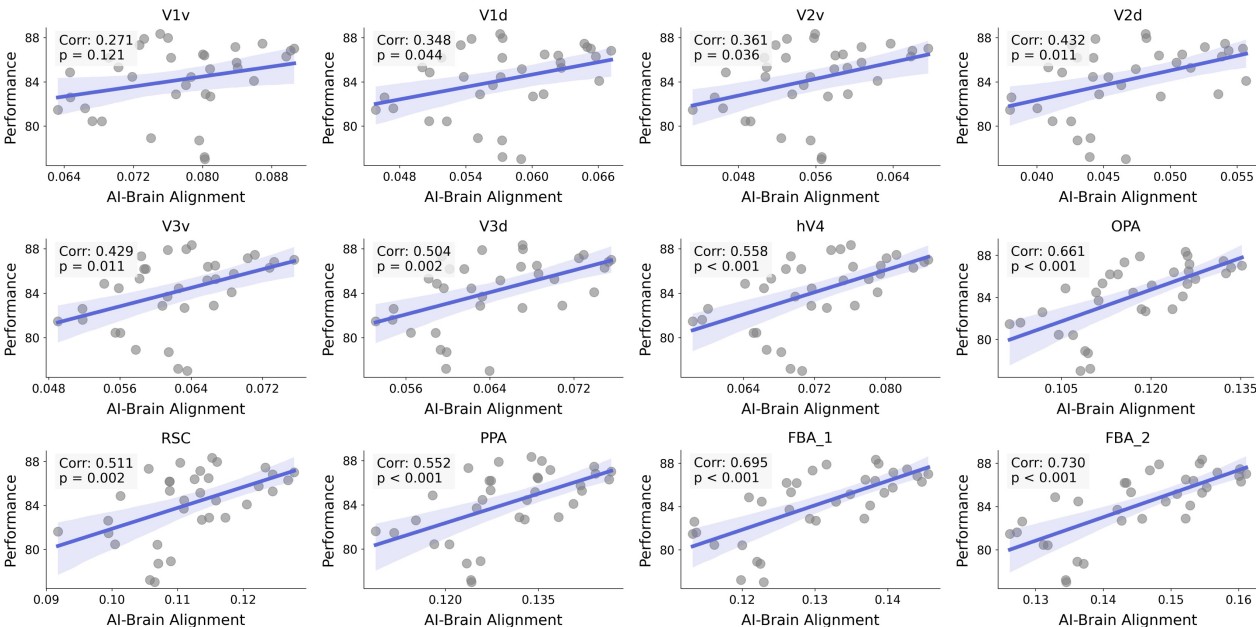

*Figure 12.* **Correlation between AI-Brain alignment and generalization performance across visual regions.** Each panel shows the relationship between AI-Brain alignment and test performance for a specific ROI. The correlation is weak and non-significant in early visual areas (e.g., V1) but increases progressively along the visual hierarchy, becoming strongest and statistically significant in higher-level visual regions.

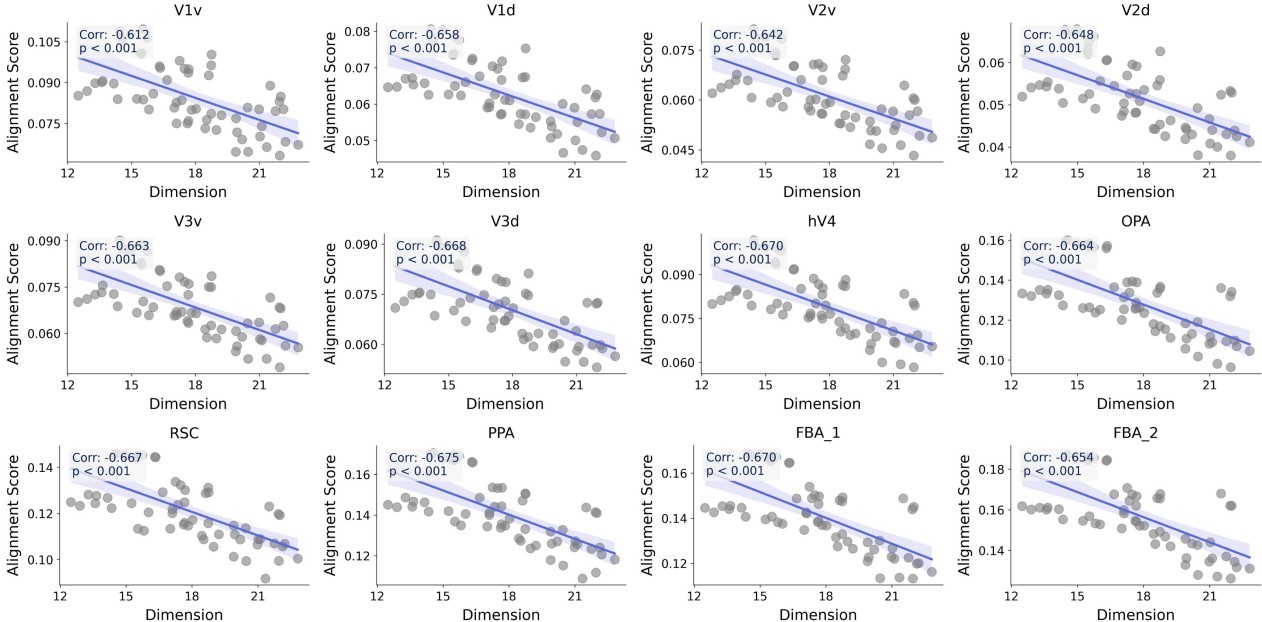

*Figure 13.* **Correlation between representational dimension and AI-Brain alignment across visual regions.** For each ROI, we report the correlation between intrinsic dimension and AI-Brain alignment across models. Significant correlations are observed in all regions, with no clear monotonic increase along the visual hierarchy.

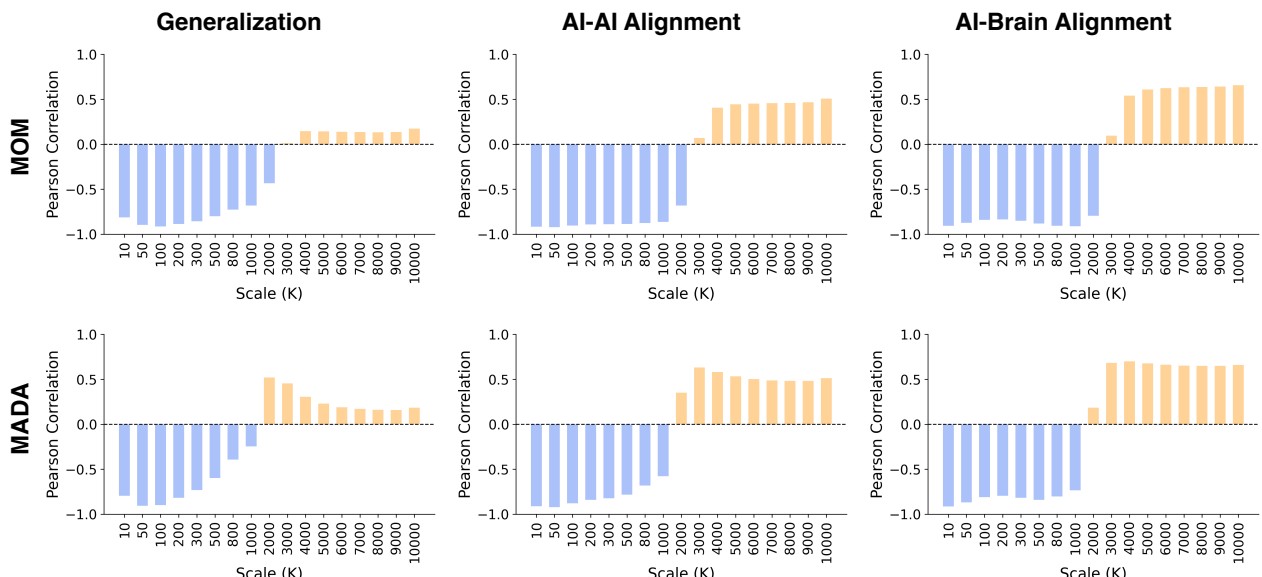

*Figure 14.* **Effect of dimension estimator choice on scale-dependent correlations.** Correlations between estimated dimension and AI-AI alignment, AI-Brain alignment, and generalization performance are shown for different estimators and neighborhood sizes. The qualitative trends are consistent across estimators, with the strongest effects occurring at small neighborhood scales.

Figure 14 shows that the overall trends are highly consistent across estimators. At larger neighborhood scales, correlations tend to weaken and may even change sign, while at smaller scales, dimension exhibits strong and significant correlations with alignment and generalization. These results indicate that the observed scale-dependent effects are not artifacts of a particular estimator but reflect intrinsic properties of the representations.

### G.2. Impact of Estimator Differences on Local Structure Analysis

Finally, we repeat the local structure analysis using alternative dimension estimators. Following the procedure in the main text, we estimate dimension based on local neighborhoods of 1,000 nearest neighbors and analyze its relationship with AI-AI alignment, AI-Brain alignment, and generalization performance.

As shown in Figure 15, all estimators yield significant correlations across a range of neighborhood sizes. The consistency across estimators further confirms that the importance of local representational geometry is robust and not driven by estimator-specific biases.

## H. Supplementary Analysis of Architecture Differences

To validate whether the findings reported in the main text generalize across different model families, we conducted additional analyses on ResNet and ResMLP architectures. Specifically, we examined inter-model alignment, model-to-fMRI alignment, generalization performance, and local embedding dimension, as shown in Figure 16.

Overall, the results are consistent across architectures. Notably, the relationship between AI-Brain Alignment and generalization performance is less pronounced in ResNet and ResMLP models. This is likely because models with higher generalization performance exhibit very similar alignment with fMRI signals (e.g., EBA $R^2$ ranging from 0.217-0.224 for ResNet, 0.175-0.285 for ResMLP), reducing variability across models.

Interestingly, ConvNeXt models achieve the highest generalization performance among all architectures. Despite ResNet and ResMLP being structurally distinct, they still exhibit significant correlations between AI-Brain Alignment, AI-AI Alignment, and generalization performance. In particular, ResNet models show that better generalization is associated with higher alignment to fMRI responses and closer embeddings to ConvNeXt models. These observations suggest that representational convergence may occur across different architectures.

Furthermore, local embedding dimension, as a geometric property, is significantly correlated with AI-AI Alignment, AI-

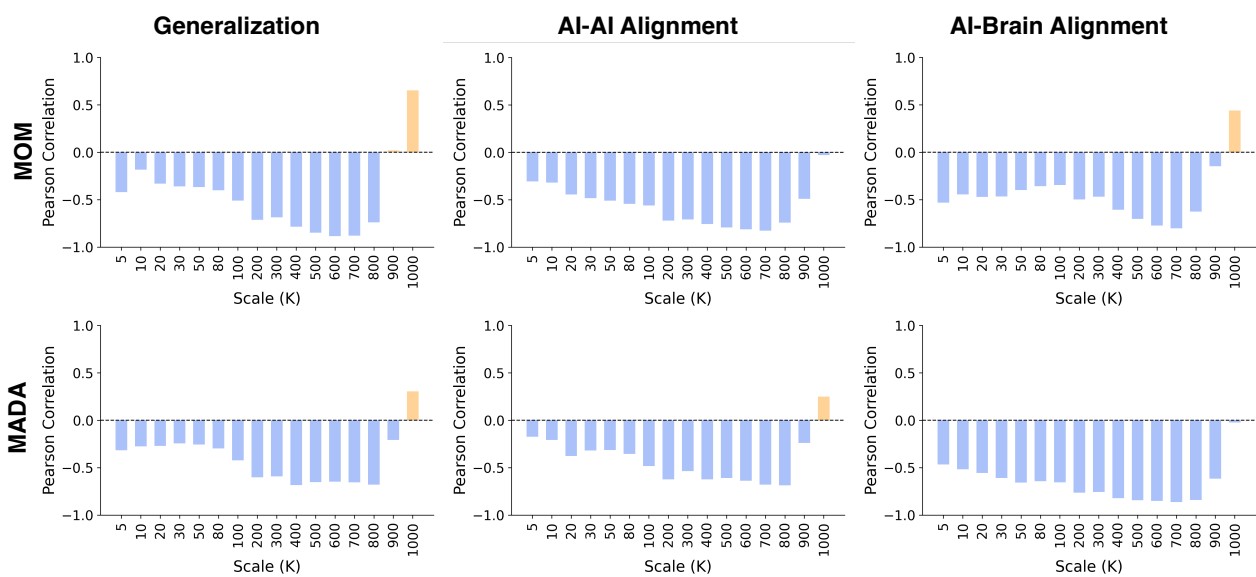

*Figure 15.* **Robustness of local structure analysis across dimension estimators.** Correlations between local intrinsic dimension and AI-AI alignment, AI-Brain alignment, and generalization performance are shown for different estimators. Significant correlations are consistently observed, supporting the robustness of the local structure findings.

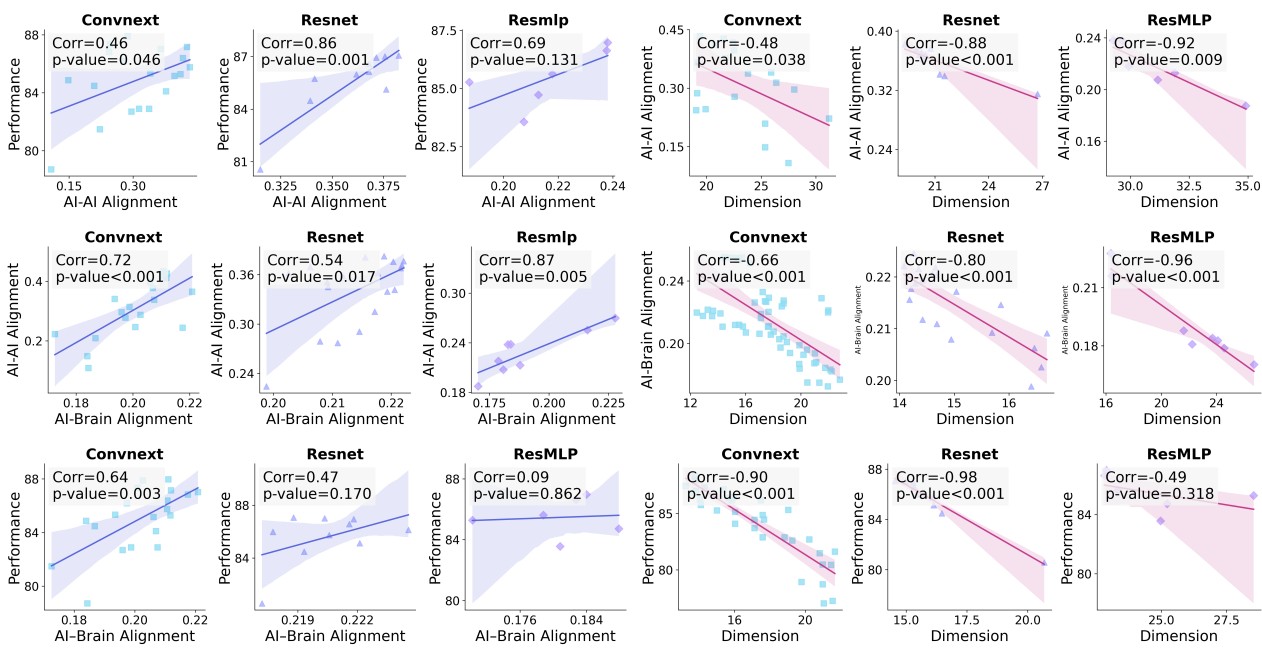

*Figure 16.* **Single-architecture analyses across ResNet, ResMLP and ConvNeXt models.** Relationships between inter-model alignment, model-to-fMRI alignment, generalization performance, and local embedding dimension are shown. Overall, trends observed in ConvNeXt are largely preserved across ResNet and ResMLP, supporting cross-architecture generality. Dimension consistently correlates with AI-AI alignment, AI-Brain alignment, and generalization performance, highlighting its importance as a geometric descriptor of model embeddings.

Brain Alignment, and generalization performance across architectures. This underscores the importance of dimension as a robust metric for characterizing model representations.

# I. Results of Language Models

The main text focuses on vision models evaluated against image-evoked fMRI responses. Because the Natural Scenes Dataset (NSD) provides only image stimuli, a direct analysis of language model–brain alignment is not feasible in our current setup.

To test whether the relationship among AI-AI alignment, generalization performance, and local intrinsic dimension extends beyond the visual domain, we repeat the analysis for a suite of pretrained language models. Table 6 summarizes the language models included in this analysis and their fine-tuned performance on the Multi-Genre Natural Language Inference (MNLI) benchmark.

*Table 6.* Pretrained language models used in the language-domain analysis and their fine-tuned accuracy on the MNLI benchmark.

| Model | MNLI Accuracy (%) |
|---|---|
| bert-base-uncased | 84.6 |
| bert-large-uncased | 86.6 |
| bert-base-cased | 84.6 |
| bert-large-cased | 86.6 |
| roberta-base | 87.6 |
| roberta-large | 90.2 |
| deberta-large | 91.3 |
| deberta-xlarge | 91.5 |
| deberta-v2-xlarge | 91.7 |
| deberta-v2-xxlarge | 91.9 |
| deberta-v3-large | 91.8 |
| bigbird-roberta-base | 87.5 |
| electra-small-discriminator | 81.6 |
| electra-base-discriminator | 88.5 |
| electra-large-discriminator | 90.9 |

Generalization is measured by accuracy on the MNLI benchmark, AI-AI alignment is computed via representational similarity across model pairs, and local intrinsic dimension is estimated from sentence embeddings using the same estimator as in the main text.

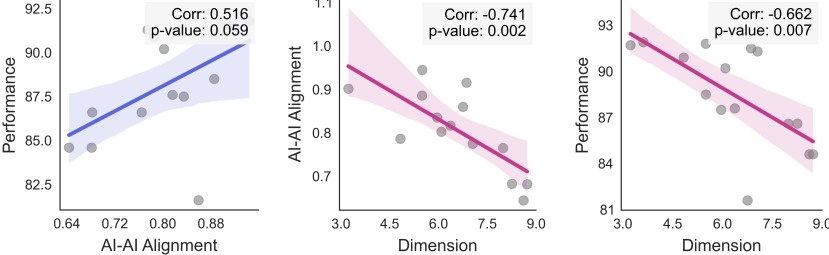

*Figure 17.* **Local intrinsic dimension predicts AI-AI alignment and generalization in language models.** Pairwise correlations among MNLI accuracy, AI-AI alignment, and local intrinsic dimension across language models. Each point represents one model. Spearman correlation coefficients and $p$-values are reported in each panel.

As shown in Figure 17, all three quantities remain significantly correlated, indicating that local intrinsic dimension serves as a domain-general geometric descriptor of representational quality.

# J. Validation of PC Dimension

In the main text, model embeddings are projected onto the top 300 principal components (PCs) prior to computing local intrinsic dimension estimates. This projection controls for differences in raw embedding dimension across architectures while retaining the dominant representational structure.

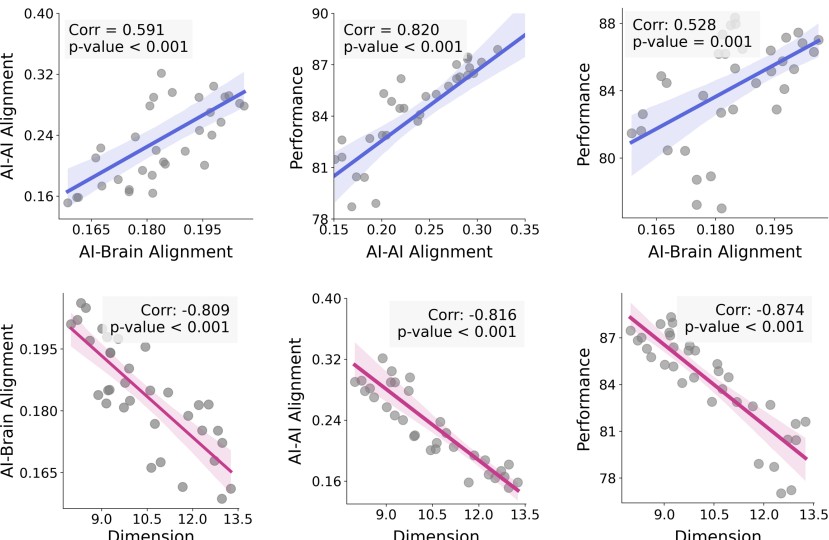

*Figure 18.* **Results are consistent when embeddings are projected onto 80 principal components.** Pairwise correlations among local intrinsic dimension, AI-AI alignment, AI-Brain alignment, and generalization, computed from embeddings projected onto the top 80 PCs. Each point represents one model. Pearman correlation coefficients and $p$-values are reported in each panel.

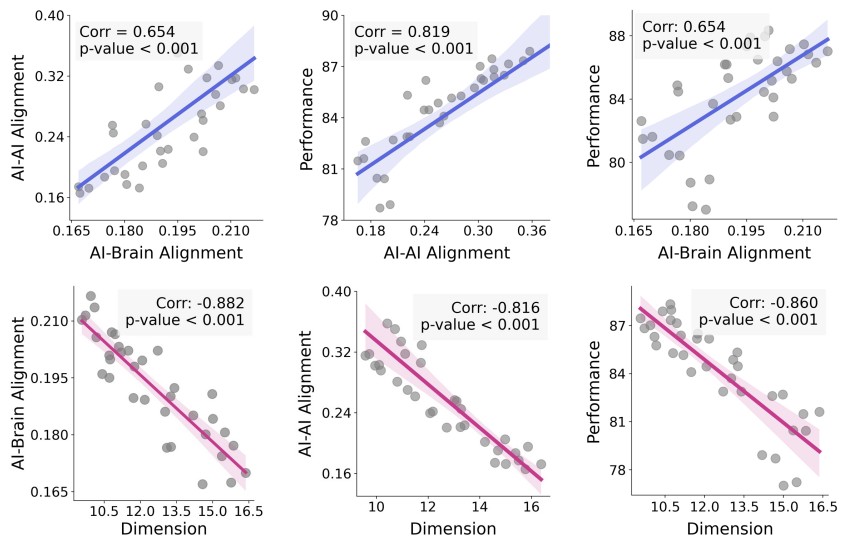

*Figure 19.* **Results are consistent when embeddings are projected onto 150 principal components.** Pairwise correlations among local intrinsic dimension, AI-AI alignment, AI-Brain alignment, and generalization, computed from embeddings projected onto the top 150 PCs. Each point represents one model. Pearman correlation coefficients and $p$-values are reported in each panel.

To verify that this choice does not qualitatively affect our conclusions, we repeat all analyses with projections onto 80 and 150 PCs respectively. As shown in Figures 18 and 19, the relationships among local intrinsic dimension, AI-AI alignment, AI-Brain alignment, and generalization remain consistent across all three projection settings, confirming robustness to the choice of PC dimension.

## K. Details of Layer-wise Analysis

The main text analyzes embeddings extracted from the final layer of each model. To investigate how representational geometry evolves across network depth, and to verify that the main findings are not exclusive to the last layer, we conduct two complementary analyses.

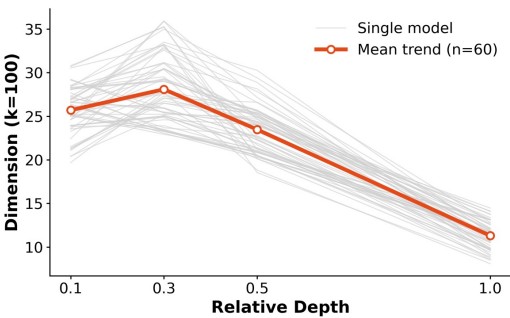

*Figure 20.* **Local intrinsic dimension decreases progressively with network depth.** Local intrinsic dimension estimated at layers corresponding to 10%, 30%, 50%, and 100% of total network depth across model families. Deeper layers consistently exhibit lower local intrinsic dimension, indicating progressive geometric compression of representations.

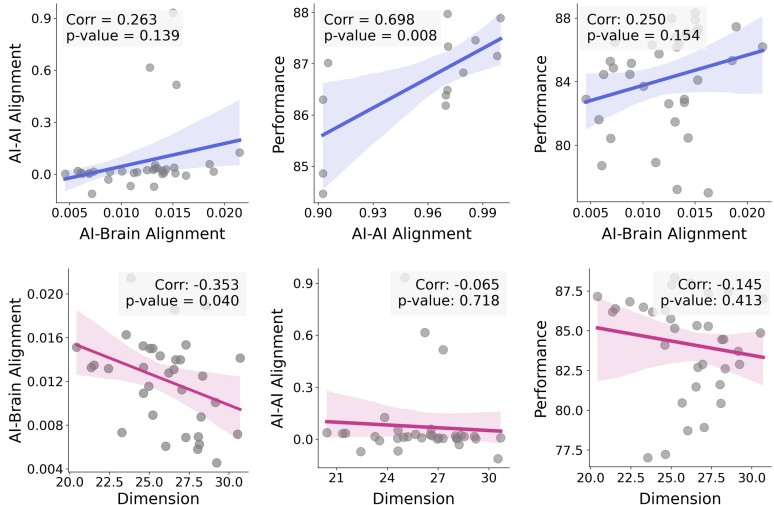

*Figure 21.* **Main findings replicate at the 10% depth layer.** Pairwise correlations among local intrinsic dimension, AI-AI alignment, AI-Brain alignment, and generalization, computed from embeddings at 10% of total network depth.

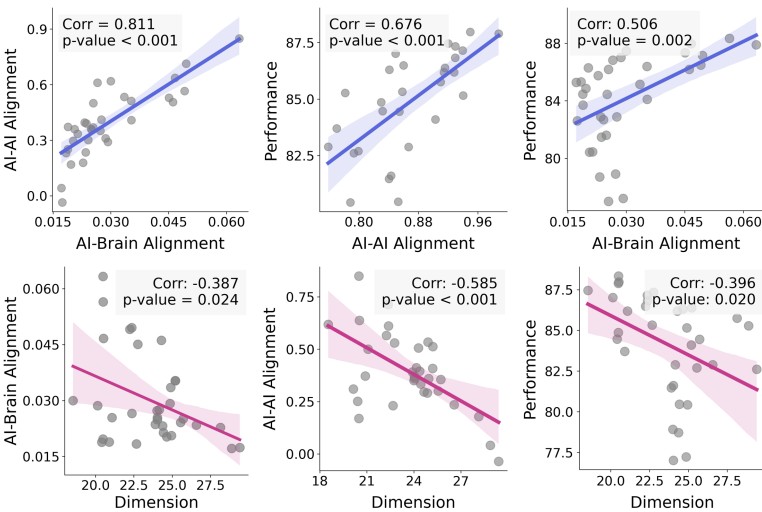

*Figure 22.* **Main findings replicate at the 50% depth layer.** Pairwise correlations among local intrinsic dimension, AI-AI alignment, AI-Brain alignment, and generalization, computed from embeddings at 50% of total network depth.

First, we track local intrinsic dimension at layers corresponding to 10%, 30%, 50%, and 100% of total network depth across all models (Figure 20). Second, we extract embeddings at the 10% and 50% depth layers and repeat the full correlation analysis among local intrinsic dimension, AI-AI alignment, AI-Brain alignment, and generalization. Since embedding dimension varies substantially across layers and architectures, directly comparing representations at their original dimensionalities could introduce confounding effects. To ensure fair comparison across models and depths, all embeddings in this layer-wise analysis are first projected to a common 100-dimensional space using PCA before computing intrinsic dimension and alignment metrics.

Results at both intermediate layers are broadly consistent with those reported in the main text, demonstrating that local intrinsic dimension remains a reliable predictor of alignment and generalization throughout the network hierarchy. Moreover, compared to representations extracted at 10% depth (Figures 21), embeddings at 50% depth exhibit stronger AI-Brain alignment and more pronounced relationships among local intrinsic dimension, AI-AI alignment, AI-Brain alignment, and generalization performance (Figures 22). This observation is consistent with the view that intermediate-to-late network layers contain increasingly abstract and behaviorally relevant representations that better capture neural response structure.

At the same time, AI-AI alignment values in these layer-wise analyses are noticeably higher than those reported in the main text. This increase should be interpreted with caution, as lower-dimensional embeddings are intrinsically easier to align across models. Because all representations are projected into a relatively low-dimensional PCA subspace prior to alignment analysis, the absolute magnitude of AI-AI alignment is inflated relative to analyses performed on higher-dimensional representations. Importantly, however, the central relationships among dimension, alignment, and generalization remain stable across layers.

## L. Incremental Predictive Power of Local Intrinsic Dimension

A potential confound in our analyses is that local intrinsic dimension may simply track model size: larger models tend to achieve better generalization and alignment, and may also exhibit lower local intrinsic dimension. To determine whether local intrinsic dimension provides explanatory power *beyond* model scale, we conduct a series of hierarchical multiple regression analyses.

For each of the three outcome variables: AI-Brain alignment, AI-AI alignment, and generalization performance (ImageNet top-1 accuracy), we first fit a baseline regression model using only $\log(\text{Params})$ as a predictor, then add local intrinsic dimension as a second predictor and record the incremental $\Delta R^2$ along with its significance.

Results are summarized in Tables 7–9. In all three cases, adding local intrinsic dimension yields a statistically significant increase in explained variance ($p - value < 0.05$ in each case), with $\Delta R^2$ ranging from 0.032 to 0.373. These results demonstrate that local intrinsic dimension captures geometrical information about model representations that is not reducible to parameter count, and independently contributes to predicting how well a model aligns with human brain responses, agrees with other models, and generalizes to held-out data.

*Table 7.* **Local intrinsic dimension provides additional explanatory power beyond model parameter count in predicting AI-Brain alignment.** Hierarchical regression results with AI-Brain alignment as the outcome variable. $\Delta R^2$ denotes the incremental variance explained by adding local intrinsic dimension (Dim) to the baseline model containing only $\log(\text{Params})$.

| Predictor(s) | $R^2$ | Adj. $R^2$ | $\Delta R^2$ | $p$ |
|---|---|---|---|---|
| $\log(\text{Params})$ | 0.389 | 0.370 | — | $8.012 \times 10^{-5}$ |
| $\log(\text{Params}) + \text{Dim}$ | 0.762 | 0.747 | 0.373 | $2.138 \times 10^{-10}$ |

*Table 8.* **Local intrinsic dimension provides additional explanatory power beyond model parameter count in predicting AI-AI alignment.** Hierarchical regression results with AI-AI alignment as the outcome variable. $\Delta R^2$ denotes the incremental variance explained by adding local intrinsic dimension (Dim) to the baseline model containing only $\log(\text{Params})$.

| Predictor(s) | $R^2$ | Adj. $R^2$ | $\Delta R^2$ | $p$ |
|---|---|---|---|---|
| $\log(\text{Params})$ | 0.598 | 0.585 | — | $1.320 \times 10^{-7}$ |
| $\log(\text{Params}) + \text{Dim}$ | 0.708 | 0.688 | 0.110 | $9.622 \times 10^{-9}$ |

*Table 9.* **Local intrinsic dimension provides additional explanatory power beyond model parameter count in predicting generalization performance.** Hierarchical regression results with ImageNet top-1 accuracy as the outcome variable. $\Delta R^2$ denotes the incremental variance explained by adding local intrinsic dimension (Dim) to the baseline model containing only $\log(\text{Params})$.

| **Predictor(s)** | $R^2$ | Adj. $R^2$ | $\Delta R^2$ | $p$ |
|---|---|---|---|---|
| $\log(\text{Params})$ | 0.901 | 0.897 | — | $1.346 \times 10^{-17}$ |
| $\log(\text{Params}) + \text{Dim}$ | 0.933 | 0.928 | 0.032 | $6.778 \times 10^{-19}$ |

