# OpenReview forum: "Local Intrinsic Dimension of Representations Predicts Alignment and Generalization in AI Models and Human Brain"
_ICML.cc/2026/Conference — ICML 2026 regular_

### Official Review · Reviewer_Mavh · 2026-02-23

**Soundness:** 2
**Presentation:** 2
**Significance:** 3
**Originality:** 3
**Overall Recommendation:** 4
**Confidence:** 3

**Summary:**

**Summary:** This paper investigates representational convergence across different artificial neural networks (ANNs), and their relationship to brain alignment and generalization. The central claim is that local intrinsic dimensionality (LID) serves as a key geometric descriptor unifying these three phenomenon -- models with lower LID tend to generalize better, align more strongly with other AI systems and align more closely with human brain representations measured via fMRI (Natural Scenes Dataset). The authors further argue that increasing model and data scale systematically reduces LID, yielding a geometric account for the empirical success of foundation models. The paper positions itself within the growing literature on the Platonic Representation Hypothesis and representational convergence (**Huh et al., 2024**; **Kornblith et al., 2019**; **Nguyen et al., 2021** -- it would be nice to add and discuss the latter two references for positioning the paper amongst broader literature).

The research question is timely and important. However, I believe that in its current form, the paper has significant issues with clarity of presentation, methodological transparency, and conceptual consistency that prevent me from recommending acceptance. I believe these issues are addressable, and I encourage the authors to revise accordingly.

**Compliance With Llm Reviewing Policy:**

Affirmed.

**Final Justification:**

I do still have some reservations about linking dimensionality with brain predictivity -- particularly, that some trivial degrees of freedom inflating alignment. If a representation space has higher local dim, it has more independent directions to along which it can vary in response to stimuli. A linear/ride regression used to predict neural responses from model activations, which is the standard brain predictivity pipeline, therefore has more "room" to find a good fit simply because there are more basis directions available to construct the mapping. In its limit, if a model's representation is full-rank and high-dimensional, a sufficiently flexible linear readout can approximate almost any target, including neural responses, regardless of whether the model has learned anything genuinely brain-like.

To put into perspective, from my understanding, if LID is high I suspect the local tangent space of the representation manifold spans more directions, giving the linear decoder more degrees of freedom to align with the neural response geometry. This implies that the correlation between LID and brain predictivity could just be a statistical artifact of the readout procedure rather than evidence of genuine representational similarity. Put differently, this model isn't more brain-like it's just easier to linearly decode

That said, I think the paper introduces some interesting ideas which would be worth debating about in the conference.

**Key Questions For Authors:**

**Minor Comments:**
The paper claims representational convergence "leaves open fundamental questions about what these shared representations look like." This framing slightly undersells prior work -- there is a rich body of literature characterizing shared representational structure (_e.g.,_ **Huth et al., 2016**; **Yamins & DiCarlo, 2016**; **Kornblith et al., 2019**). The authors should be more precise about what specifically is not yet understood that this paper addresses. For instance, the connection to explanatory models in neuroscience (**Kaplan & Craver, 2011**) could be made more explicit if the authors wish to make normative claims about what a good model of brain representations should look like.

**Presentation and Figure clarity.** I found several of the figures difficult to interpret.
1. **Figure 2C:** It is unclear which models are being compared and what "inter-model variability" refers to in this context. Perhaps the caption/main text could make this explicit.
2. **Figure 2D:** The heatmap would be considerably more interpretable if model names (or at least architecture families and parameter counts) were labeled along the axes. Without this it is difficult to assess whether, for example, parameter count is confounding the alignment patterns.
3. **Figure 4:** The meaning of "global" and "local" scales $(K)$ is not explained in the main text at the point where **Figure 4** is referenced. This should be defined clearly before the figure is introduced.
4. **Figure 5:** Multiple issues arise here. First, what does each point in the scatter plots represent -- a layer, a model, a subject? The referenced passage (**L377**–**L380**) discusses AI-brain and AI-AI alignment, but it is not clear how to read off both from the same figure. Second, Figures **5A** and **5B** are both labeled "AI-brain alignment" -- what is the distinction between them? Third, for the inter-model heatmap in Figure **5G**, it is not clear whether each cell represents a single layer pair or an aggregation across layers.
5. **Some confusion in Figures 5G/H.** The discussion around Figures **5G** and **5H** contains what appears to be a tension that warrants clarification. On one hand, the paper argues that lower LID is associated with stronger AI-AI alignment. On the other hand, the authors note in the context of Figure **5H** that cross-architecture alignment is lower, but that average LID does not differ systematically across architectures, concluding that “the reduced cross-architecture alignment cannot be explained by dimensionality differences alone.” These two claims are not necessarily contradictory, but the relationship between them is not clearly articulated. Is the claim that LID is _sufficient_ to explain within-architecture alignment but not _necessary_ for cross-architecture differences? Or is LID one of several relevant factors? The authors may have a coherent account of this, but it is not currently communicated clearly.

**References**:
1. M. Huh, B. Cheung, T. Wang, and P. Isola. The platonic representation hypothesis (2024)
2. S. Kornblith, M. Norouzi, H. Lee, and G. Hinton. Similarity of neural network representations revisited (2019)
3. T. Nguyen, M. Raghu, S. Kornblith. Do Wide and Deep Networks Learn the Same Things? Uncovering How Neural Network Representations Vary with Width and Depth (2021)
4. Ansuini, A., Laio, A., Macke, J. H., and Zoccolan, D. Intrinsic dimension of data representations in deep neural networks (2019)
5. Sharma, U. and Kaplan, J. Scaling laws from the data manifold dimension (2022)
6. R. Cao and D. Yamins. Explanatory models in neuroscience: Part 2–constraint-based intelligibility (2021)
7. C. Kapoor, S. Srivastava, M. Khosla. Bridging Critical Gaps in Convergent Learning: How Representational Alignment Evolves Across Layers, Training, and Distribution Shifts (2025)

**Limitations:**

Yes

**Strengths And Weaknesses:**

**Strengths:**
1. **Unified framework**. The paper's most compelling contribution is the attempt to establish a _single_ geometric descriptor (LID), that jointly is predictive of AI-AI and AI-brain alignment as well as generalization performance. If substantiated, I believe that this could be a meaningful step beyond prior work that treats this phenomenon separately.
2. **Multi-scale analysis of intrinsic dimensionality.** The finding that _local_ rather than global intrinsic dimensionality is a more predictive quantity is a very non-trivial and interesting result. It adds nuance to some prior work -- linking representational dimensionality and generalization (**Ansuini et al., 2019**; **Sharma & Kaplan, 2022**) which have largely relied on global geometric structure or near-global estimates.
3. **Scaling estimates.** The mechanistic account of why scale reduces LID, and why this correlates with downstream performance is a useful stepping stone contribution to gaining better understanding of foundational models from a geometric perspective.
4. **Connection to existing theory.** The findings are broadly consistent with, and could perhaps be framed as empirical support for the “contravariance principle” in **Cao & Yamins (2024)**, which posits that harder or more diverse tasks constrain the space of viable representational solutions on a smaller manifold, hence driving convergence across models and brains. The papers findings that LID correlates with stronger alignment and better generalization could perhaps be interpreted as identifying a local geometric signature of this kind of constraint – representations that occupy compressed local neighbourhoods may be reflective of smaller solution space imposed by far richer training objectives. The authors do not currently draw this connection, but doing so would situate this contribution with existing theory and provide a principled account of _why_ LID should be the relevant descriptor.

**Weaknesses:**
1. **Conceptual framing.** The paper would benefit from a more principled _a priori_ motivation for why intrinsic dimensionality is expected to be the “right” descriptor of representational convergence, rather than a post-hoc framing around an empirical finding. Is the intuition here that better-generalizing models learn to compress local neighborhoods of the data manifold more efficiently? If so, this should be stated clearly and connected to existing theory. In particular, the relationship to the contravariance principle (as stated earlier) seems relevant – lower-dimensional representations in the sense used here may correspond to a more “invariant” geometry that is favored by both task optimization and biological constraints. Making this connection explicit would substantially strengthen the conceptual contribution and help situate the paper within the broader literature on why brains and models converge. In a similar vein, the authors describe LID as a "task-agnostic" descriptor of representational geometry, but this claim is not well-justified. LID is computed from the representational geometry itself, which is shaped by the training task. In what sense is it task-agnostic? Is it that the value of LID transfers across tasks, or that it is predictive of alignment regardless of which task alignment is evaluated on? This needs more clarification.
2. **Experimental Details: NSD Dataset.** While the manuscript performs extensive experiments on all subjects from NSD (Appendix **G**) for the brain alignment analyses, I was wondering if the large inter-subject variability could be a byproduct of the fact that of the eight subjects, only four (subjects $1, 2, 5, 7$) share a common set of $1000$ images could be a confound here?
3. **Experimental Details: Layer Selection.** While this is by no means a high priority experiment, previous work has shown substantive evidence supporting that representational geometry varies substantially across layers (**Kapoor et al., 2025**), and the choice of layer could drive the observed results. Could it be possible to examine the effect of, say, picking early, middle and late convolutional layers towards intrinsic dimensionality? I would suspect that LID would be lower in early layers and gradually increase with model depth given the fact that early layers learn domain-general filters, which progressively becomes task-specific.

---

> ### Author Rebuttal · Authors · 2026-03-29
>
> We thank the reviewer for the thoughtful comments. The added experiments are also available at the anonymous link: https://anonymous.4open.science/r/2026_ICML_Rebuttal-CF9C/README.md
>
> **1. Conceptual framing and "task-agnostic" terminology**
>
> Our choice of LID is not post hoc, but follows from two a priori criteria.
>
> * First, since prior work shows that better-generalizing models tend to align better with the brain (Yamins et al., 2014; Schrimpf et al., 2018, 2021), we sought a representation-level quantity already linked to generalization, with the expectation that it could also help explain alignment.
>
> * Second, we wanted a measure applicable across both supervised and self-supervised models without relying on labels, so that comparisons remain fair across training paradigms. This rules out label-dependent measures such as separability.
>
> Dimension satisfies both criteria: it has known links to generalization  (Ansuini et al., 2019), can be computed directly from embeddings, and has a natural geometric interpretation in terms of compression. We therefore selected dimension as a principled candidate for relating alignment and generalization.
>
> On the contravariance principle: We agree that this is broadly consistent with the contravariance principle and will make this connection more explicit.
>
> On "task-agnostic": We agree the term is imprecise, since dimension is shaped by the training task, what we mean is that it is annotation-free, and we will revise the wording accordingly.
>
> **2. NSD dataset and inter-subject variability**
>
> In our analysis, AI-brain alignment is computed separately for each subject using that subject’s own full image set (10,000 images in NSD), rather than the 1,000-image subset shared by subjects. Therefore, the presence of a shared stimulus subset should not confound the within-subject alignment estimates or explain the variability across subjects.
>
> Similar inter-subject variability has also been reported in prior work on NSD (e.g., Wang et al., 2023, Fig. S2), suggesting that it is a known property of the dataset. Importantly, despite these differences, our main correlations remain positive and significant for every individual subject (Appendix E.2–E.4).
>
> **3. Layer selection and depth dependence**
>
> We conducted a systematic layer-depth analysis by extracting embeddings from layers at approximately 10%, 30%, 50%, and 90% of the model depth. We found that dimension first increases and then decreases with depth (**Figure 5 at the link**), consistent with Fig. 3B of Ansuini et al. (2019). AI-brain alignment is weaker in shallow layers and stronger in deeper layers, and the correlations of dimensionality with both alignment and generalization likewise strengthen with depth (**Figure 6 and 7 at the link**).
>
> **4. Clarifying the gap addressed**
>
> Our work addresses two specific gaps.
>
> * First, prior studies have examined AI-AI alignment and AI-brain alignment, *but they relied on different models, datasets, and analysis metrics, making direct comparison difficult*. Our work is the first to systematically analyze AI-AI alignment, AI-brain alignment, and generalization within a unified framework.
>
> * Second, while representational geometry has been studied in relation to brain function (e.g., Huth et al., 2016; Yamins & DiCarlo, 2016) and model generalization, it remains unclear *which properties are associated with the degree of AI-brain alignment*. Here, for the first time, we show that dimension is jointly related to both alignment and generalization, and that local dimensionality is more informative than global dimensionality.
>
> **5. Figure clarity**
>
> * Figure 2C. Each box shows the distribution of model-wise mean AI-brain alignment scores for a given ROI, wider boxes indicate greater inter-model variability.
>
> * Figure 2D. We have added a figure at the anonymous link (**Figure 8**) that labels the full model names and AI-brain alignment scores. Parameter is indeed significantly correlated with alignment (Figure 6 of the paper), but multivariate regression shows that LID explains additional variance in AI-AI alignment, AI-brain alignment, and generalization beyond parameter (**Table1-3 in the anonymous link**).
>
> * Figure 4. K is defined as scale in Section 3.4 (line 143). Larger K captures more global representational structure, smaller K reflects local geometry. We will add this to the caption.
>
> * Figure 5. Each point represents one model. Figures 5A and 5B both use AI-brain alignment but examine different relationships: 5A plots it against AI-AI alignment, 5B against generalization. In Figure 5G, each cell shows the alignment between the final-layer embeddings of a pair of models.
>
> * Figures 5G–H. Our intended claim is that LID is sufficient to account for within-architecture alignment trends, but not for cross-architecture differences, which are additionally shaped by architecture-specific biases. We will clarify this in the text.

---

> > ### Author Rebuttal · Reviewer_Mavh · 2026-04-02
> >
> > I sincerely thank the authors for patiently addressing my questions! I also apologize for having overlooked certain obvious answers to my questions in the Figure panels/descriptions -- thanks for this. All my concerns have now been addressed and I will raise my score accordingly.

---

> > > ### Author Response · Authors · 2026-04-05
> > >
> > > Thank you again for your thoughtful feedback and kind follow-up. We sincerely appreciate your suggestions, which have helped us improve the paper substantially. In particular, we will revise the motivation to make the conceptual framing clearer, and we will further improve the figure captions and related presentation so that the paper is easier to follow for readers. Your comments have greatly helped strengthen the quality and clarity of our work.

---

### Official Review · Reviewer_ebHZ · 2026-03-06

**Soundness:** 3
**Presentation:** 3
**Significance:** 3
**Originality:** 3
**Overall Recommendation:** 4
**Confidence:** 2

**Summary:**

This paper studies AI-AI, AI-brain alignment, and their relationship to generalisation. They show that models with stronger generalisation also align strongly with human data. The paper further investigates how both the local and global dimensions of the embedding correlate with the generalisation performance. In my view, the strongest contribution is the link between intrinsic local dimension and generalisation.

**Compliance With Llm Reviewing Policy:**

Affirmed.

**Final Justification:**

My concerns have been adequately addressed (See rebuttal acknowledgement).

**Key Questions For Authors:**

I have a few questions to clarify some of the concerns highlighted above and better understand the paper's limitations and contributions.

1. Could you more precisely described your contributions in the context of the citations provided above?
2. Could you further detail the FMRI data set?
3. Do you see a similar trend with amount of data during training humans ( I am not sure if the dataset allows this) ?
4. How do you interpret the fact that the AI-Brain alignment increase with the number of parameters?

I look forward to your answer and remain open to increasing my score upon clarification of the contributions and limitations.

**Limitations:**

The paper points out several interesting limitations. The interpretation of the alignment is an important limitation to further study.

However, important limitations are left out. The paper does not discuss that this universal alignment is contested (Dujmovic et al. 2022,Conwell et al., 2022; Soni et al., 2024, Klabunde et al, 2024). These papers point out the importance of the similarity measure and data structure in the alignment.

**Strengths And Weaknesses:**

I would like to first thank the author for their effort and for this interesting work.

**Soundness**

**Strengths:** The paper is technically sound, and the claims are well supported by experimental analysis.


**Presentation**

**Strengths:** The paper is written in good English and has a good structure. The figures help the understanding, and are clear.

**Weaknesses:** However, one downside is that the motivation, intro, and related work sections are too broad, and the paper is not well positioned in the current literature, which creates clarity issues. I discuss this more in the significance park. It would be also helpfull to further detail the benchmark used ( number of participant etc)


**Significance**

**Strengths:** The paper is interesting, timely, and highly relevant to the ICML community.  This is highlighted by the fact that, in the last few years, multiple workshops on this topic (unifying representations across brain and machines) have been organised at NeuRIPS and ICML ( UniReps, ReAlign). This is an important direction for future research (intersection between Ai and Neuroscience) .

**Weaknesses:**
 The gap this paper fills is too broad, and I point to some literature below that I believe could provide more context.

1. **On the link between alignment and generalisation:** Moshella et al (2022) have shown a correlation between alignment and generalisation performance. It would also be important to cite this paper and its related literature. Furthermore, the claim  that 'AI-AI alignment and AI-brain alignment have been studied separately' does not fully make sense to me. The field of AI-Brian alignment examines whether AI and Brian representations align with the same thing, which would be equivalent to AI-AI alignment. My understanding of the contribution is that you show that models with stronger generalisation also align strongly with human data. Therefore, I would like to ask if you could better describe your contribution, ‘Unified Framework: We establish the strong correlation between AI-AI alignment, AI-Brain alignment, and Generalisation.'

2. **On the geometric principle:**  The paper shows interesting results on the relationship between dimensionality and generalisation.  However, other papers have characterised these representations geometrically (Chung et al. 2021) and linking them to generalisation, notably in the field of neuroscience (Fusi et al., 2016; Neil et al., 2023).  The literature on the rich-and-lazy learning regime also links the model's representation and geometry to its generalisation (Braun et al, 2025, Dominé et al 2024, Chizat et al. 2019, Kunin et al. 2024, Woodworth et al. 2020).  I understand that these papers are not directly in the subfield if the paper, but I believe they bring context to the geometric insight and tools developed in this paper. Here again, reframing the contribution would be important.

Due to the lack of strong positioning in the literature, it is hard for me to assess the contribution.

**Originality**

**Strengths:**
This paper offers new insights that bridge neuroscience and AI. It is original in linking the alignment with brain representation to the model's generalisation. Furthermore, its investigation of the characteristics of the representation can be considered original
and geometrical insights

**Weaknesses:**
This is provided that the paper clarifies its contributions.

---

> ### Author Rebuttal · Authors · 2026-03-29
>
> Thank you for these thoughtful comments.
>
> **1. Clarifying our contribution relative to prior work**
>
> ***Contribution 1: jointly analyzing AI-AI alignment, AI-brain alignment, and generalization under a unified evaluation framework.***
>
> Prior work has already established several relevant pieces of the picture.
>
> * On AI-AI alignment, studies such as Huh et al. (2024) showed that stronger models tend to align better with one another.
>
> * On the AI-brain alignment, prior work showed that better-performing models often better predict neural activity (e.g., Yamins et al., 2014; Schrimpf et al., 2018, 2021). Other studies further showed that model-level factors such as task design, and the scale or diversity of training data can substantially affect brain alignment (e.g., Wang et al., 2023; Conwell et al., 2024).
>
> However, these findings have mostly been studied separately and often at the model level, using different model sets, datasets, and evaluation protocols. Our contribution is therefore not to pose these questions for the first time, but to study them jointly within the same experimental framework. Specifically, we evaluate alignment and generalization on the same set of models, using matched stimuli and a shared analysis pipeline, and show that models with stronger generalization systematically exhibit both stronger inter-model alignment and stronger alignment to fMRI.
>
> ***Contribution 2: local intrinsic dimension as a shared geometric correlate.***
>
> We agree that representation geometry is not a new topic. Prior work in neuroscience and machine learning has characterized neural or model representations geometrically and related geometry to function or generalization (Fusi et al., 2016; Chung et al. 2021; Moshella et al., 2022).
>
>
> Our contribution is not to introduce geometry as a new perspective per se, but to show that, in this joint setting, a single representational property, local intrinsic dimension, tracks AI-AI alignment, AI-brain alignment, and generalization simultaneously. Moreover, we find that local dimension is more informative than global dimension for explaining these relationships in our setting. We will revise the manuscript to clarify this positioning.
>
>
> **2. Clarifying the fMRI benchmark**
>
> We use the Natural Scenes Dataset (NSD), which includes 8 participants scanned with 7T fMRI while viewing natural scene images largely drawn from Microsoft COCO. The dataset contains approximately 73,000 images in total. Each participant viewed about 9,000 to 10,000 images, including 1,000 shared images, and contributed about 22,000 to 30,000 trials. The fMRI data have about 1.8 mm isotropic resolution. We will clarify these details in the main text and appendix.
>
> **3. On the question about training data in humans**
>
> We see two possible interpretations.
>
> If it refers to AI training data, our results suggest that larger training scale is associated with better generalization, stronger AI-brain alignment, and lower local intrinsic dimensionality, as shown in Figure 6B.
>
> If it refers to human learning over time, NSD does not support that analysis because it is a stimulus-response dataset rather than a longitudinal one. We therefore cannot test whether increasing human experience leads to stronger alignment or systematic dimensionality changes. This would be an interesting future direction with suitable longitudinal neural data.
>
> **4. Interpreting the increase of AI-brain alignment with model size**
>
> We do not interpret this pattern as showing that parameter count itself directly makes a model more brain-like. Rather, larger models are typically trained at a larger overall scale, often with more data, which provides stronger constraints on the learned representation. Under these constraints, the model is less able to rely on idiosyncratic or non-robust features and is instead driven to retain structure that is shared across many examples and relevant to the task. We believe this selective compression of irrelevant variability is one plausible mechanism behind the observed reduction in local intrinsic dimensionality. In turn, such lower-dimensional and more robust representations may align better with brain responses.
>
>
> **5. On limitations and contested interpretations of “universal alignment”**
>
> We thank the reviewer for this important point. We agree that stronger interpretations of “universal alignment” should be treated cautiously, as prior work has shown that alignment can depend substantially on the similarity measure and dataset structure, and that high alignment alone does not imply shared mechanisms between AI models and the brain. We will revise the manuscript to cite this literature explicitly and discuss these issues more clearly.

---

> > ### Author Rebuttal · Reviewer_ebHZ · 2026-04-02
> >
> > I thank the authors for their thoughtful and thorough response.
> >
> > Regarding Contribution 2 (local intrinsic dimension as a shared geometric correlate), I apologize if my earlier question was unclear; I was referring to the latter point.
> >
> > I encourage the authors to incorporate these clarifications into the manuscript, as doing so would further enhance its clarity and overall quality. In particular, strengthening the positioning within the existing literature and more explicitly articulating the addressed gap (also noted by other reviewers e.g., Reviewer Mavh) would be especially valuable.

---

> > > ### Author Response · Authors · 2026-04-05
> > >
> > > Thank you for the clarification and constructive suggestion. We agree that the manuscript would benefit from a clearer positioning within the existing literature and a more explicit statement of the gap addressed by our work. We will revise the paper accordingly by strengthening the related work discussion and sharpening the presentation of our contributions.

---

### Official Review · Reviewer_9rGG · 2026-03-12

**Soundness:** 3
**Presentation:** 3
**Significance:** 2
**Originality:** 2
**Overall Recommendation:** 5
**Confidence:** 4

**Summary:**

This paper jointly analyzes AI-AI alignment, AI-Brain alignment, and generalization performance across a large set of vision models and human fMRI data (Natural Scenes Dataset). The core finding is that these three measures are strongly correlated with each other, and that a single geometric property, the local intrinsic dimensionality of learned representations, predicts all three. Through a multi-scale analysis, the authors show that dimensionality estimated at local (small neighborhood) scales is far more predictive than global estimates. They further show that increasing model capacity or training data size systematically reduces local intrinsic dimensionality, offering a geometric lens on scaling laws. The paper includes extensive robustness checks across subjects, brain regions, architectures (ConvNeXt, ResNet, ResMLP, ViT), and dimensionality estimators.

**Compliance With Llm Reviewing Policy:**

Affirmed.

**Key Questions For Authors:**

How sensitive are results to the PCA threshold of 300? Have you tried 100 or 1000 components?

Have you tested on OOD benchmarks like ImageNet-V2 or ObjectNet?

**Limitations:**

The restriction to vision models is acknowledged in Section 5, as is the lack of theoretical mechanism. The authors could be more upfront about the purely correlational nature of the findings and the narrow definition of generalization.

**Strengths And Weaknesses:**

Strength 1: The biggest contribution is bringing AI-AI alignment, AI-Brain alignment, and generalization into a single analysis (Section 4.1, Figure 2). Prior work looked at these in isolation. The correlations are strong (r=0.722, 0.770, 0.816 in Figure 2F) and the reference-based alignment measure is carefully constructed to avoid the circularity that would come from using a brain-optimal reference.

Strength 2: The multi-scale analysis in Section 4.3 is really well done. Showing that correlations peak at small K (Figure 4C), persist within fixed local neighborhoods (Figure 4D), and weaken under random global subsampling at matched sample size (Figure 4E) is convincing evidence that local structure is what matters. This goes meaningfully beyond prior work that estimated dimensionality at a single scale.

Strength 3: The robustness checks are extensive and genuinely useful. All 8 NSD subjects show consistent patterns (Appendix E, Figures 8-10), the effects hold across brain regions from V1 through higher visual areas (Appendix F), three different dimensionality estimators give consistent results (Appendix G), and the trends generalize across architecture families (Appendix H, Figure 16). This is the kind of supplementary material that actually builds confidence.

Weakness 1: The paper frames its findings as explanatory when they're correlational. Throughout Sections 4.2, 4.5, and 5, language like "captures," "explains," and "geometric account" is used, but there's no intervention or causal analysis. Model quality (training procedure, data volume, capacity) could independently drive dimensionality, alignment, and generalization all at once. The flat minima discussion in Section 5 is interesting but entirely speculative since no loss landscape measurements are provided. The paper would be stronger if it consistently used "predicts" or "is associated with" rather than implying mechanism.

Weakness 2: The scope is narrower than the claims suggest. Everything is restricted to vision models, ImageNet-1K evaluation, and visual cortex fMRI data. No language models, no multimodal models beyond CLIP-trained ConvNeXts, no other tasks or domains. But the title says "AI Models and Human Brain" and the conclusion talks about "artificial and biological systems" broadly. Section 5 acknowledges this as a limitation in passing, but this should be made more apparent

Weakness 3: "Generalization performance" always means ImageNet-1K top-1 accuracy (Sections 4.1, 4.2), which is a pretty narrow proxy. It conflates architecture design, training data, augmentation strategies, and actual generalization ability. Testing on out-of-distribution benchmarks like ImageNet-V2 or ObjectNet would make the generalization claims much more convincing.

Weakness 4: Intrinsic dimensionality is estimated on the PCA-reduced embeddings, not the original space (Sections 3.2, 3.4). PCA is a linear compression that could alter local geometric structure. It's unclear whether the local-vs-global distinction in Section 4.3 would survive if dimensionality were estimated in the original high-dimensional space.

---

> ### Author Rebuttal · Authors · 2026-03-29
>
> Thank you for these helpful comments. We have revised the manuscript to better align the claims with the actual scope and evidence. The newly added experiments are also available at the following anonymous link: https://anonymous.4open.science/r/2026_ICML_Rebuttal-CF9C/README.md
>
> **1. On the concern that some of our language was explanatory rather than correlational.**
>
> We agree with the reviewer. Our evidence is correlational rather than causal, and we have revised the manuscript accordingly. In particular, we systematically replaced causal or mechanistic wording such as “explains” and “captures” with more accurate expressions such as “predicts” and “is associated with”.
>
> We also clarified the role of the flat minima discussion in Section 5. Its purpose is to provide a possible theoretical interpretation and a hypothesis for future work, not to present an established mechanism. We will make this distinction explicit.
>
> **2. On the concern that the scope is narrower than some of the claims suggest.**
>
> We agree that the original wording was too broad. We have revised the claims to match the actual empirical scope of the paper, which is currently limited to vision models, ImageNet-based evaluation, and visual cortex fMRI. In particular, we removed broad phrasing such as “artificial and biological systems” where it could overstate the generality of the results.
>
> Because NSD is based on visual stimuli, it cannot be used to assess alignment between language models and fMRI responses. Nevertheless, to test whether our geometric findings generalize beyond vision, we collected a new set of language models and analyzed the relationship among AI-AI alignment, embedding dimensionality, and generalization performance on MNLI (**Figure 1 of the anonymous link**). We observed the same statistically significant trend in language models, suggesting that the geometry–alignment–generalization relationship is not unique to vision.
>
>
> **3. On the concern that “generalization performance” is too narrowly defined by ImageNet-1K top-1 accuracy.**
>
> We agree that, in the current version of the paper, performance is primarily defined based on ImageNet-1K accuracy, and we will revise the manuscript to make this point more precise.
>
> In addition, we have now collected a new set of models for which ImageNet-V2 performance is available and re-ran our analyses. The updated results show that under this new metric, the relationships among AI-AI alignment, AI-brain alignment, generalization performance, and embedding dimension remain consistent with the main findings of the paper. At the same time, we note that the fMRI alignment results for this new set of models are relatively similar across models, which makes some of the corresponding effects less statistically significant. These new results are shown in **Figure 2 at the anonymous link**.
>
> **4. On estimating intrinsic dimensionality after PCA reduction rather than in the original embedding space.**
>
> We first apply PCA to the embeddings of all models before analysis in order to avoid confounding effects introduced by differences in embedding width across models. Controlling for this factor is necessary not only for dimensionality estimation, but also for other analyses such as ridge regression, where differences in raw feature dimensionality could otherwise affect the results. By projecting all models into a common PCA space, we improve cross-model comparability and reduce the influence of architecture-specific embedding size.
>
> To address the reviewer’s concern about sensitivity to the PCA threshold, we added additional analyses with different numbers of principal components, including 80 and 150 (**Figure 3 and 4 at the anonymous link**). We could not use 1000 components because some models have much smaller embedding widths, which would break comparability across models. The main relationships among AI-AI alignment, AI-brain alignment, ImageNet-1K performance, and dimensionality remain consistent across these settings. We will clarify this rationale and report the sensitivity analysis in the revision.

---

> > ### Author Rebuttal · Reviewer_9rGG · 2026-04-01
> >
> > I am satisfied with the rebuttal/extra experiments provided by the authors and have updated my score accordingly.

---

> > > ### Author Response · Authors · 2026-04-04
> > >
> > > Thank you for your thoughtful and constructive feedback throughout the review process. We are glad that the rebuttal and additional experiments addressed your concerns, and we will carefully incorporate all of your suggestions into the final revision of the manuscript.

---

### Official Review · Reviewer_XDJ8 · 2026-03-13

**Soundness:** 3
**Presentation:** 3
**Significance:** 3
**Originality:** 3
**Overall Recommendation:** 5
**Confidence:** 3

**Summary:**

The paper investigates representational convergence between artificial vision models and the human brain through the lens of local intrinsic dimension. The authors establish that generalization performance, model-model alignment, and model-brain alignment (fMRI) are significantly correlated with one another. They argue that these relationships are explained by the geometric complexity of learned representations: lower local intrinsic dimension is consistently associated with stronger alignment and better generalization. Finally, the study provides a geometric account of scaling laws, demonstrating that increasing model capacity and training data systematically reduces local intrinsic dimensionality, thereby driving representational convergence.

**Compliance With Llm Reviewing Policy:**

Affirmed.

**Final Justification:**

I find both the motivations and methodology of this paper of interest and high impact for the field, which is why i grade it 5.

**Key Questions For Authors:**

- Can you provide a deeper theoretical or geometric account of the causal mechanism by which increasing model capacity and training data scale drives representations into lower-dimensional local neighborhoods, rather than just documenting the correlation?

- Could you clarify the results in Figure 4C, specifically addressing the observed inversion of Pearson correlation and how it reconciles with the finding that larger model scales simultaneously elicit higher correlation for AI-Brain alignment?

- Causal Mechanisms: Can you elaborate on why "increasing model capacity and training data scale systematically reduces local intrinsic dimension"? Is this a byproduct of the optimization landscape (e.g., flat minima) or an inherent property of natural data manifolds?.

- Generality Across Modalities: You noted that "extending this framework to other modalities, such as language or audition, will be important." Do you have preliminary evidence that local dimension predicts alignment in LLMs or speech models?.


- Spatial Scale K: In your multi-scale analysis (Figure 4), you find local dimensionality is most predictive. Is there a specific physical or biological interpretation for the optimal neighborhood size K used in your estimators?.

**Limitations:**

The authors adequately discuss that their analysis is currently restricted to vision models and visual cortex responses. They also acknowledge that while local intrinsic dimensionality strongly predicts alignment and generalization, the theoretical mechanisms underlying this relationship remain unclear and are a subject for future research.

**Strengths And Weaknesses:**

**Soundness**

* Strength: The authors perform a robust multi-scale analysis, demonstrating that local dimensionality is a more sensitive and informative indicator of model quality than global measures.


* Strength: The experimental design carefully controls for dimensionality differences by using a standardized PCA projection onto the top 300 principal components across all models before alignment scoring.


* Weakness: While the negative correlation between local dimension and performance is strong (e.g., $r = -0.916$ in Figure 3D), the theoretical causal mechanism—why exactly scaling drives representations into lower-dimensional local neighborhoods—remains speculative.


* Weakness: There is an unclear result in Figure 4C: why is their an inversion of pearson correlation? And for AI-Brain alignment, bigger scale also seem to elicit bigger Pearson correlation.


**Presentation**

* Strength: The paper is logically structured, moving from broad alignment phenomena to specific geometric explanations and scaling laws.


* Strength: Figure 1 effectively illustrates the motivation and overview, linking convergent representations to locally low-dimensional spaces.


* Weakness: For the reliance on ridge regression for the "alignment score", the authors need to cite established literature for this method.



* Weakness: The manuscript retains the default "Submission and Formatting Instructions for ICML 2026" header on multiple pages, which detracts from its professional presentation.


**Significance**

* Strength: The study unifies AI-AI alignment and AI-Brain alignment under a single geometric principle, addressing a major gap in the "neuro-AI" literature.


* Strength: Identifying local dimensionality as a descriptor for representational "quality" offers a practical, task-agnostic tool for evaluating foundation models without requiring explicit neural data.


* Weakness: The findings are currently restricted to the visual modality, focusing primarily on vision models and human visual cortex responses.


* Weakness: The literature review in Introduction regarding how scaling models increases brain alignment is incomplete; the authors should complete their review with recent complementary findings, eg "Scaling and context steer LLMs along the same computational path as the human brain".



**Originality**

* Strength: The work introduces a novel geometric account for scaling laws, suggesting that the benefits of scale arise from a progressive refinement of local geometric structure.


* Strength: It moves beyond documenting the existence of alignment to providing a mathematical language—Local Intrinsic Dimensionality—to describe the convergent representations themselves.

---

> ### Author Rebuttal · Authors · 2026-03-29
>
> Thank you for the careful reading and constructive feedback. The newly added experiments are also available at the following anonymous link: https://anonymous.4open.science/r/2026_ICML_Rebuttal-CF9C/README.md
>
>
> **1. The mechanism behind the negative correlation between local intrinsic dimension and performance.**
>
> We agree that our current results support correlation rather than causation, and we will revise the manuscript to avoid causal language.
>
> A plausible interpretation is that scaling promotes more invariant and more constrained representations. In natural vision, generalization requires collapsing nuisance variation while preserving task-relevant structure. With limited data, many solutions remain compatible with the task, including representations that retain additional nuisance-sensitive local variability. As data and model scale increase, representations must remain stable across a wider range of nuisance conditions, which places stronger pressure on the model to compress irrelevant variation and organize embeddings around shared task-relevant factors. Geometrically, this would manifest as fewer locally variable directions, i.e., lower local intrinsic dimensionality. We emphasize that this is a hypothesis consistent with our findings, not a causal mechanism established by the present study.
>
> **2. The apparent inversion of Pearson correlation in Figure 4C and its relation to larger model scale improving AI-brain alignment.**
>
> We first clarify a terminology issue: in Figure 4C, “scale” refers to the neighborhood size K used in LID estimation (i.e., geometric scale), not model or data scale. We will revise the text to make this distinction explicit.
>
> Regarding the inversion of the Pearson correlation in Figure 4C, we interpret it as reflecting a change in what LID captures as the neighborhood size K increases. When K is small, the estimate is dominated by genuinely local structure and is more likely to characterize within-class (or within-cluster) variability. In this regime, lower LID indicates that nearby samples vary along fewer task-relevant directions, consistent with more stable local geometry.
>
> As K increases, the neighborhood increasingly spans multiple clusters or classes, so LID becomes sensitive not only to local manifold structure but also to between-class arrangement. In this regime, lower dimension can arise when samples are more tightly concentrated around class centers, reflecting stronger cluster structure rather than beneficial local invariance. Such large-scale cluster structure may help category separation, but it is less consistent with the brain’s more continuous coding of visual information (Huth et al., 2012). This may explain why, at larger K, lower dimension no longer implies better AI-Brain alignment and can even reverse its correlation.
>
> This is not contradictory, because “model scale” and “LID scale” refer to different quantities. Larger models improve AI-Brain alignment overall, while Figure 4C shows that this relationship is best captured by small-K local geometry rather than larger-K estimates.
>
> **3. A more complete literature review for both the alignment methodology and recent scaling-related brain-alignment findings.**
>
> Thank you for pointing this out. We will add appropriate references for both the ridge-regression-based alignment score and recent complementary work on scaling and AI–brain alignment.
>
> **4. Presentation and formatting.**
>
> We have corrected the formatting issue and removed the unintended header.
>
> **5. The scope of the findings and whether they generalize beyond vision.**
>
> We agree that our current empirical results are limited to the visual modality, and we will state this limitation more clearly.
>
> Since NSD is a dataset of image stimuli, we are unable to analyze the alignment between language models and fMRI in this context. However, we leveraged a new set of language models to examine the relationships among embedding dimension, inter-model alignment, and generalization performance on the MNLI dataset, showing that these factors are significantly correlated. The results are presented in ***Figure 1*** at the anonymous link.
>
>
> **6. The interpretation of the neighborhood size K.**
>
> We view K primarily as a methodological scale parameter rather than a quantity with direct biological meaning. Small K probes local geometry, while large K increasingly reflects global structure.
>
> The existence of an optimal K reflects a standard trade-off: when K is too small, the estimate becomes noisy due to limited samples; when K is too large, it no longer captures local manifold structure.
>
> Therefore, the optimal K should be interpreted as the point that best balances estimation stability and geometric locality, rather than a biological constant.

---

> > ### Author Rebuttal · Reviewer_XDJ8 · 2026-04-03
> >
> > Thank you for having taken the time for this strong rebuttal, even with an initial high score. Most of my concerns have been addressed. Could you please provide your list of recent complementary work on scaling and AI–brain alignment ?

---

> > > ### Author Response · Authors · 2026-04-04
> > >
> > > Thank you for the follow-up. Below is the list of recent complementary works we plan to incorporate into the revised introduction.
> > >
> > > **AI–Brain Alignment**
> > >
> > > * Goldstein et al. showed that contextual embeddings from large language models share common geometric patterns with neural activity recorded from inferior frontal gyrus, suggesting a convergent representational structure between language models and the human brain [1].
> > >
> > > * Yu et al. demonstrated that adding next sentence prediction to masked language modeling better captures discourse-level neural responses, particularly in regions associated with high-­level language understanding [2].
> > >
> > > **Scaling and Alignment**
> > >
> > > * Gokce et al. demonstrated that neural alignment with the primate visual ventral stream saturates as model and dataset size scale up, while increasing training data consistently yields greater alignment gains than expanding model parameters [3].
> > >
> > > * Raugel et al. systematically disentangled the contributions of model scale, training data size, and data domain to AI-brain alignment. They found that models trained on human-centric images, as opposed to satellite or biological imagery, achieve significantly better AI-brain alignment, underscoring the joint importance of model scale and data composition [4].
> > >
> > > * Raugel et al. demonstrate that LLM representations across layers align with the temporal dynamics of human brain responses during language processing, and that both model scale and context length are key factors underlying this AI-brain alignment [5].
> > >
> > > **References**
> > >
> > > [1] Goldstein, Ariel, et al. "Alignment of brain embeddings and artificial contextual embeddings in natural language points to common geometric patterns." Nature communications 15.1 (2024): 2768.
> > >
> > > [2] Yu, Shaoyun, et al. "Predicting the next sentence (not word) in large language models: What model-brain alignment tells us about discourse comprehension." Science advances 10.21 (2024): eadn7744.
> > >
> > > [3] Gokce, Abdulkadir, and Martin Schrimpf. "Scaling Laws for Task-Optimized Models of the Primate Visual Ventral Stream." International Conference on Machine Learning. PMLR, 2025.
> > >
> > > [4] Raugel, Joséphine, et al. "Disentangling the factors of convergence between brains and computer vision models." arXiv preprint arXiv:2508.18226 (2025).
> > >
> > > [5] Raugel, Joséphine, et al. "Scaling and context steer LLMs along the same computational path as the human brain." arXiv preprint arXiv:2512.01591 (2025).

---

### Decision · Program_Chairs · 2026-04-30

**Decision:**

Accept (regular)

**Comment:**

This paper makes a solid and timely contribution by jointly analyzing AI-AI alignment, AI-brain alignment, and generalization within a unified framework, and by identifying local intrinsic dimension as a useful geometric correlate of these quantities across a broad set of vision models and brain-alignment analyses. Reviewers generally found the empirical study technically strong, with especially positive feedback on the multi-scale dimensionality analysis, the breadth of robustness checks, and the additional experiments and clarifications provided in the rebuttal.

The main concerns centered on scope, framing, and presentation rather than on core soundness. In particular, reviewers noted that the claims should remain clearly correlational rather than causal, that the paper’s scope should be stated more carefully given that the main evidence is in vision and visual cortex data, and that the work should be better situated within the existing literature on representational alignment, geometry, and related brain-model comparison frameworks. The authors’ rebuttal addressed these issues constructively, including clarifying the scope of the claims, adding sensitivity analyses and additional experiments, and committing to revise the manuscript accordingly.

On balance, I recommend accepting. For the camera-ready version, the authors should make the changes suggested by the reviewers, especially strengthening the positioning within the representational alignment literature, clarifying the specific gap relative to prior work, softening any causal or overly broad claims, and incorporating the requested presentation and citation improvements.